# IN-CONTEXT LEARNING OF TEMPORAL POINT PROCESSES WITH FOUNDATION INFERENCE MODELS

**David Berghaus**[1,2]**, Patrick Seifner**[1,3]**, Kostadin Cvejoski**[4]**,**
**César Ojeda**[5] & **Ramsés J. Sánchez**[1,2,3]
Lamarr Institute[1], Fraunhofer IAIS[2], University of Bonn[3], JetBrains Research[4]
& University of Potsdam[5]
`david.berghaus@iais.fraunhofer.de`

## ABSTRACT

Modeling multi-type event sequences with marked temporal point processes (MTPPs) provides a principled framework for uncovering governing dynamical rules and predicting future events. Current neural approaches to MTPP inference typically require training separate, specialized models for each target system. We pursue a fundamentally different strategy: leveraging amortized inference and in-context learning, we pretrain a deep neural network to infer, *in-context*, the conditional intensity functions of event histories from a context consisting of sets of event sequences. Pretraining is performed on a large synthetic dataset of MTPPs sampled from a broad distribution over point processes. Once pretrained, our `Foundation Inference Model for Point Processes` (`FIM-PP`) can estimate MTPPs from real-world data without additional training, or be rapidly finetuned to specific target systems. Across common benchmark datasets, `FIM-PP` matches the performance of specialized models in *zero-shot mode*. After only a few finetuning iterations, `FIM-PP` further improves its predictions and *outperforms competing methods on the majority of evaluated tasks*.

Our pretrained model, repository, and tutorials are available online[1]

## 1 INTRODUCTION

The mathematical modeling of asynchronous, irregular event sequences has long occupied a distinctive place in the machine learning community. Temporal point processes provide the canonical framework for modeling neural dynamics (Truccolo et al., 2005; Linderman & Adams, 2014), and they serve as the de facto tool for describing a wide range of internet phenomena, including retweeting, posting, and information cascades (Zhao et al., 2015; Cvejoski et al., 2020). Their ability to encode fine-grained temporal structure, together with their capacity to reveal causal interactions between events in an interpretable manner, has made them indispensable not only in neuroscience and social media, but also in finance (Aït-Sahalia et al., 2015) and epidemiology (Chiang et al., 2022). Despite this centrality, the development of foundation models has followed a different trajectory. Large-scale pretraining first emerged in natural language processing, enabled by massive internet corpora, and has only recently been extended to dynamical systems, with recent work addressing ODEs (d'Ascoli et al., 2024; Mauel et al., 2025; 2026), MJPs (Berghaus et al., 2024), SDEs (Seifner et al., 2025a), and even specific applications in pharmacology (Marin et al., 2025). It is therefore ironic that event data — the very modality underlying the internet activity that made text-based pretraining possible — has not yet given rise to a corresponding foundation model for point processes. The present work takes a first step toward filling this gap by developing a foundation model explicitly designed for temporal point processes.

Marked Temporal Point Processes (MTPPs) (Daley & Vere-Jones, 2007; Rasmussen, 2018) are stochastic processes consisting of ordered occurrence times, each accompanied by a categorical mark specifying its type. Formally, the objective is to specify the conditional distribution of the next event time and mark given the history of the process up to the current time. The extensive point

---

[1] `https://fim4science.github.io/OpenFIM/intro.html`

process literature explores different ways of encoding event histories and specifying the stochastic mechanism that governs new arrivals and their marks (Lin et al., 2024). A common approach is to represent this distribution through a *conditional intensity function*, which describes the instantaneous rate at which events of different types occur given the history. Traditionally, models such as the Hawkes process (Hawkes, 1971) define this conditional intensity as a superposition of self-exciting effects from past events. Forecasting is then carried out recursively using Ogata's thinning algorithm, applied to the conditional intensity. Building on this cornerstone, more recent work has extended Hawkes processes by parameterizing event histories with neural architectures (Mei & Eisner, 2017), including recurrent neural networks (Du et al., 2016), attention mechanisms (Zhang et al., 2020), and Transformers (Zuo et al., 2021). These models are typically trained either via maximum likelihood — often requiring expensive integral evaluations — or through generative approaches that bypass intensity modeling altogether and directly sample future events conditional on the past (Kerrigan et al., 2024; Zeng et al., 2024). However, a fundamental limitation of these approaches is their *lack of transferability*: each new dataset requires retraining from scratch, forcing the model to relearn representations for each distinct dynamical regime.

In contrast, modern approaches to dynamical systems increasingly prioritize *pretraining on synthetic data*, yielding general models that can learn dynamics *in-context*. This paradigm offers a crucial advantage: practitioners no longer need to train models *de novo* for every dataset, but can instead obtain accurate characterizations in a *zero-shot* manner. Within this family, two variants have emerged: Prior-fitted Networks (PFNs) and Foundation Inference Models (FIMs). PFNs train networks to approximate *predictive posterior distributions* in a sequence-to-sequence or context-to-sequence fashion (Müller et al., 2022; Hollmann et al., 2022; Müller et al., 2025). FIMs, by contrast, focus on directly *estimating the infinitesimal operators* of dynamical system (e.g., drift and diffusion functions for SDEs), thereby retaining a degree of interpretability (Berghaus et al., 2024; Seifner et al., 2025a;b; Mauel et al., 2026). Access to these operators enables the explicit study of physically relevant observables such as entropy production, stationary distributions, and attractors.

In the context of point processes, we adapt the FIM pretraining paradigm to MTPPs in three steps. First, we define a broad family of conditional intensity functions, thereby inducing a diverse prior over MTPPs. This prior captures assumptions about excitatory and inhibitory effects between events, as well as the interaction structure across event types. Second, we sample MTPPs from this prior, generate synthetic event sequences, and construct tuples consisting of context event sequences, event histories, and their corresponding intensities, thereby creating a meta-learning task that amortizes inference across heterogeneous dynamics. Third, we train a neural network to recover conditional intensities from observed context. We summarize our contributions as follows:

1. Introduce a synthetic data generation framework for sampling event sequences from a *broad prior distribution over MTPPs*, with randomized base intensities, kernels, and interaction types (i.e., excitatory, inhibitory, neutral). We empirically demonstrate that this construction encodes a strong prior, enabling models trained on it to *generalize* across both in-distribution processes and real-world event data.

2. Train the first transformer-based recognition model capable of estimating *in-context* the conditional intensity functions of MTPPs, where history representations serve as queries and the encoded sequence context provides the keys and values.

3. Show that the resulting model achieves strong *zero-shot* performance across synthetic benchmarks and multiple real-world datasets, and that it can be *rapidly finetuned on new event data*.

## 2 RELATED WORK

Here we provide a brief overview of temporal point processes. For detailed surveys and benchmarks of deep TPP models, including open challenges in history encoding, conditional intensity design, relational discovery, and learning strategies, see e.g., Lin et al. (2024) and Xue et al. (2024).

While the mathematical theory of point processes is extensive (Daley & Vere-Jones, 2007; Kingman, 1992), work on temporal point processes (TPPs) in machine learning has crystallized around two central questions: (i) how should representations of past events be constructed, and (ii) how should the future be modeled (Lin et al., 2024). Early approaches, epitomized by the Hawkes process,

addressed both questions using linear self-exciting kernels. A natural extension is the neural Hawkes process (Mei & Eisner, 2017), along with related recurrent formulations (Du et al., 2016), which rely on neural representations of past events, trained via likelihood maximization, and model future events using the thinning algorithm. Later work introduced more expressive architectures. Attention mechanisms (Zuo et al., 2021; Zhang et al., 2020; Yang et al., 2021) extend the memory horizon of TPPs, though at the cost of higher computational demand. Neural ODEs (Chen et al., 2018) have also been incorporated to better capture the irregular timing of events in latent representations (Song et al., 2024; Kidger et al., 2020).

To improve predictive accuracy over long horizons, different decompositions of the likelihood for future arrivals have been proposed (Rasmussen, 2018; Panos, 2024; Deshpande et al., 2021; Draxler et al., 2025). These works highlight the limitations of intensity-based inference, particularly when relying on thinning algorithms. Such limitations have motivated a shift toward generative models, which typically sample entire sequences. Approaches include optimal transport (Xiao et al., 2017), diffusion models (Zeng et al., 2024; Lüdke et al., 2023), and flow-matching methods (Kerrigan et al., 2024), often trading accuracy for interpretability. In contrast, traditional machine learning methods (Rasmussen, 2013; Malem-Shinitski et al., 2022) emphasize interpretability from the outset. A key advantage of the Hawkes process is that its excitation graph makes causal structure explicit (Xu et al., 2016; Wu et al., 2024), which has been especially relevant in neuroscience (Linderman & Adams, 2014; Truccolo et al., 2005) and in finance, where Hawkes models often serve as hidden drivers of observed activity (Aït-Sahalia et al., 2015). Applications extend more broadly, for instance to dynamics of text (Cvejoski et al., 2020; 2021), social online activity (Zhao et al., 2015) and operations research (Ojeda et al., 2021).

While Liu & Quan (2024) fine-tuned a pre-trained LLM for point-processes (with modest success), to the best of our knowledge, no prior work has presented a foundation model for point-processes that can operate zero-shot without further training.

## 3 PRELIMINARIES

In this section, we recall the definition and basic properties of *marked temporal point processes* (Daley & Vere-Jones, 2007) and *Hawkes processes* (Hawkes, 1971; Laub et al., 2015). Additionally, we define the inference problem our proposed approach tackles.

**Marked Temporal Point Processes:** We consider *marked temporal point processes* (marked TPPs, or MTPPs) as simple point processes on $\mathbb{R}_+ \times \mathcal{K}$, where $\mathcal{K}$ is a discrete and finite set of *marks*. The density $f$ of a sequence of events $\mathcal{S} = \{(t_i, \kappa_i)\}_{i=1}^n$ in the interval $[0, T]$, w.l.o.g. ordered by their *time component* $t_i \in \mathbb{R}_+$, factors into *conditional densities*

$$f\left(\{(t_i, \kappa_i)\}_{i=1}^n\right) = \prod_{i=1}^n f\left((t_i, \kappa_i) \mid \mathcal{H}_{t_i}\right) = \prod_{i=1}^n f(t_i \mid \mathcal{H}_{t_i}) f(\kappa_i \mid t_i, \mathcal{H}_{t_i}), \tag{1}$$

where $\mathcal{H}_t = \{(t_i, \kappa_i) \mid t_i < t\} \subset \mathcal{S}$ is the *history strictly preceding* $t$. By the last equality of equation 1, MTPPs may be characterized by dependent densities of the *next-event time* $f(t \mid \mathcal{H}_t)$ and its *event mark* $f(\kappa \mid t, \mathcal{H}_t)$. MTPPs are commonly represented by their piece-wise continuous *conditional intensity function*

$$\lambda(t, \kappa \mid \mathcal{H}_t) = \frac{f(t \mid \mathcal{H}_t)}{1 - \int_{t'}^t f(s \mid \mathcal{H}_s) ds} f(\kappa \mid t, \mathcal{H}_t) = \lambda(t \mid \mathcal{H}_t) f(\kappa \mid t, \mathcal{H}_t), \tag{2}$$

where $t'$ is the last event time in $\mathcal{H}_t$, or $t' = 0$ if $\mathcal{H}_t = \varnothing$. The conditional intensity function may be interpreted of the *instantaneous rate* of mark $\kappa$ occurring at $t$, conditioned of the history up to time $t_i$. Reversely, any such function $\lambda$, satisfying some mild conditions, defines the density of an MTPP on a set of events in an interval $[0, T]$ by

$$f\left(\{(t_i, \kappa_i)\}_{i=1}^n\right) = \left[\prod_{i=1}^n \lambda(t_i, \kappa_i \mid \mathcal{H}_{t_i})\right] \exp\left(-\int_0^T \lambda(s \mid \mathcal{H}_s) ds\right). \tag{3}$$

**Collection of TPPs:** A TPP is just an MTPP with a single mark. An MTPP can be viewed as a *collection of TPPs* per mark, interdependent through a *joined history*. Indeed, given an MTPP as

above, the conditional intensity $\lambda_\kappa(t \mid \mathcal{H}_t) = \lambda(t \mid \mathcal{H}_t) f(\kappa \mid t, \mathcal{H}_t)$ defines the *marginal TPP* for mark $\kappa \in \mathcal{K}$, that may depend on other marks via $\mathcal{H}_t$. Conversely, a collection of TPPs with conditional intensity $\lambda_\kappa$ per $\kappa \in \mathcal{K}$ can be *joined* to an MTPP. Using

$$\lambda(t \mid \mathcal{H}_t) = \sum_{\kappa \in \mathcal{K}} \lambda_\kappa(t \mid \mathcal{H}_t) \quad \text{and} \quad f(\kappa \mid t, \mathcal{H}_t) = \frac{\lambda_\kappa(t \mid \mathcal{H}_t)}{\lambda(t \mid \mathcal{H}_t)}, \tag{4}$$

in Eq. 2 defines the conditional intensity function $\lambda(t, \kappa \mid \mathcal{H}_t)$ of an MTPP. In fact, $\lambda(t, \kappa \mid \mathcal{H}_t) = \lambda_\kappa(t \mid \mathcal{H}_t)$. In contrast to some other neural methods (Du et al., 2016), which estimate $\lambda(t \mid \mathcal{H}_t)$ and $f(\kappa \mid t, \mathcal{H}_t)$, we design our model to parametrize TPPs per mark, conditioned on the joined history of all marks.

**Hawkes Processes:** A *Hawkes MTPP* with marks $\mathcal{K}$ is defined by the conditional intensity

$$\lambda(t, \kappa \mid \mathcal{H}_t) = \max\left(0, \mu_\kappa(t) + \sum_{(t', \kappa') \in \mathcal{H}_t} \gamma_{\kappa\kappa'}(t - t')\right), \tag{5}$$

where $\{\mu_k\}_{\kappa \in \mathcal{K}}$ is a set of *time-dependent base intensity functions*, and $\{\gamma_{\kappa\kappa'}\}_{\kappa, \kappa' \in \mathcal{K}}$ is a set of *interaction kernels*, specifying the influence of mark $\kappa'$ on mark $\kappa$. If $\gamma_{\kappa\kappa'}$ is positive, the influence of $\kappa'$ on $\kappa$ is called *excitatory* or *exciting*, otherwise it is called *inhibitory* or *limiting*.

**Simulation:** We use *Ogata's modified thinning algorithm* (Ogata, 1981) to generate synthetic training data from MTPP, and to simulate processes inferred by our model.

**Inference Problem:** Let $\mathcal{C} = \{\mathcal{S}^j\}_{j=1}^m$ be a collection of $m$ event sequences $\mathcal{S}^j = \{(t_i^j, \kappa_i^j)\}_{i=1}^{n_j}$ observed from some system. Our objective is to *predict* or *simulate* the *next event* and estimate the *likelihood* of a (previously unseen) sequence $\mathcal{S}$, assuming an MTPP model. Previous neural methods *train an autoregressive encoding network on $\mathcal{C}$ that compresses the history $\mathcal{H}_t$ of $\mathcal{S}$ into some embedding $\mathbf{h}_t$ for a neural estimate $\hat{\lambda}(t, \kappa \mid \mathbf{h}_t)$ of the conditional intensity. In contrast, we *pretrain* a deep neural network model to estimate *in-context* $\hat{\lambda}$, given a history of events, from a collection of context event sequences. Once trained, the model can be applied to *any* $\mathcal{C}$ and $\mathcal{H}_t$, *without any further training*.

## 4 Foundation Inference Models for Point Processes

In this section, we present a novel *in-context learning method* for the MTPP intensity inference problem. In a two-step approach, we first generate a *large set of marked event sequences* from parametrized MTPPs, sampled from a *broad distribution* over MTPPs. This yields train data for a neural network recognition model, trained to estimate the *underlying known, ground-truth* intensity functions. Such *pretrained* inference model can be applied directly to real-world problems, or *swiftly finetuned* for improved performance.

### 4.1 Synthetic Dataset Generation

To construct a synthetic dataset of MTPPs, we define a distribution over MTPPs via a distribution over conditional intensity functions of the form

$$\lambda(t, \kappa \mid \mathcal{H}_t) = \max\left(0, \mu_\kappa(t) + \sum_{(t', \kappa') \in \mathcal{H}_t} z_{\kappa\kappa'} \gamma_{\kappa\kappa'}(t - t')\right). \tag{6}$$

Given a sample from this distribution, we simulate a large set of marked event sequences and record the corresponding conditional intensities for training.

We use Eq. 6 to parameterize several classes of processes:

- The classical Hawkes process, for which $\mu_k$ is constant in time and $\gamma_{\kappa\kappa'}(t)$ is an exponential function;
- The Poisson process, for which $\mu_k$ is constant in time and $\gamma_{\kappa\kappa'} = 0$;

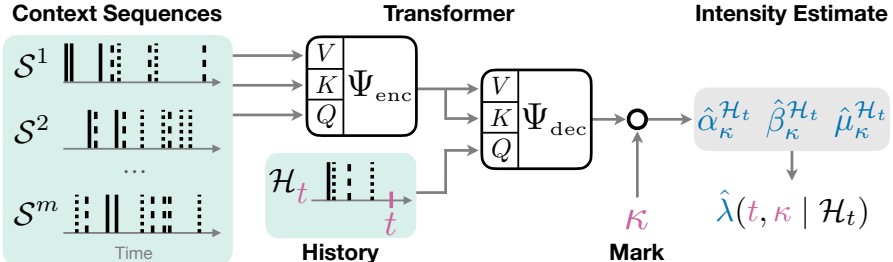

Figure 1: Schematic representation of `FIM-PP`. A *context* of marked event sequences $\mathcal{S}^j$ is encoded by a self-attentive *transformer encoder*. The result is further processed by a *transformer decoder*, using a *history* $\mathcal{H}_t$ of marked events before time $t$ as queries. The final embedding is joined with an encoding of *mark* $\kappa$. The results is projected to a set of parameters that determine the value of the *conditional intensity function* $\hat{\lambda}$ evaluated at $(t, \kappa)$.

- Periodic processes, for which $\mu_k(t)$ is a positive sinusoidal function;
- Processes with high initial excitation, for which we choose $\mu_k(t)$ to follow a Gamma distribution;
- Processes with non-monotonic, shifted kernels, for which we choose $\gamma_{\kappa\kappa'}(t)$ to follow a Rayleigh distribution.

A complete overview of the process families and their hyperparameters is provided in Table 6 in the Appendix. Furthermore, Figure 7 in the Appendix reports summary statistics of our simulated pretraining distribution, and confirms that it is broad enough to cover the real-world datasets depicted in Figures 8 to 12, also in the Appendix.

For each process, we randomly sample $z_{\kappa\kappa'} \in \{-1, 0, 1\}$ for $\kappa, \kappa' \in \mathcal{K}$ to cover *excitatory* ($z_{\kappa\kappa'} = 1$), *inhibitory* ($z_{\kappa\kappa'} = -1$) and *non-influencing* ($z_{\kappa\kappa'} = 0$) interactions.

### 4.2 FOUNDATION INFERENCE MODEL ARCHITECTURE

We now present the architecture of `FIM-PP`, a pretrained deep neural network for inference of MTPPs from sets $\mathcal{C} = \{\mathcal{S}^j\}_{j=1}^m$ of marked event sequences $\mathcal{S}^j = \{(t_i^j, \kappa_i^j)\}_{i=1}^{n_j}$. `FIM-PP` processes the *context sequences* $\mathcal{C}$ and estimates the conditional intensity function $\hat{\lambda}$ of an MTPP that describes the observed dynamics. Following previous intensity-based methods (Zhang et al., 2020; Zuo et al., 2021), $\hat{\lambda}$ is implemented by a flexible parametrized function family $\hat{\lambda}(\cdot, \kappa \mid \mathcal{H}_\cdot)$ for all $\kappa \in \mathcal{K}$. `FIM-PP` estimates its parameters by encoding the history $\mathcal{H}_t = \{(t_i^{\text{hist}}, \kappa_i^{\text{hist}})\}_{i=1}^{n_{\text{hist}}}$ before time $t > t_{n_{\text{hist}}}^{\text{hist}}$, *subject to the processed context sequences*. Figure 1 depicts a schematic representation of this approach.

To cover applications in different time scales, `FIM-PP` instance normalizes its inputs and renormalizes $\hat{\lambda}$ accordingly. Appendix C provides the details. Once trained, `FIM-PP` can be applied for all counts of marks $|\mathcal{K}|$ up to some fixed upper bound, similar to in-context methods in other domains (d'Ascoli et al., 2024; Berghaus et al., 2024).

We denote linear projections by $\phi$, feed-forward neural networks by $\Phi$, attention layers with residual connections by $\psi$, transformer encoders by $\Psi_{\text{enc}}$ and decoders by $\Psi_{\text{dec}}$. Let $E \in \mathbb{N}$ denote the model's embedding dimension.

**Context Encoding:** To encode $\mathcal{C}$, we combine encodings of individual sequences $\mathcal{S}^j$. Recognizing the importance of inter-observation times for the inference problem, we consider $\Delta t_i^j = t_i^j - t_{i-1}^j$ as an additional feature, identifying $t_0^j = 0$. To encode $\mathcal{S}^j$, we first embed the features $(t_i^j, \kappa_i^j, \Delta t_i^j)$ of the $i$-th event in sequence $j$ into embeddings

$$\mathbf{u}_i^j = \phi_t(t_i^j) + \phi_\kappa(\kappa_i^j) + \phi_{\Delta t}(\Delta t_i^j) \in \mathbb{R}^E . \tag{7}$$

Sinusoidal output activations from Shukla & Marlin (2020) enhance the networks $\phi_t$ and $\phi_{\Delta t}$. Let $\mathbf{S}^j = [\mathbf{u}_1^j, \ldots, \mathbf{u}_{n_j}^j] \in \mathbb{R}^{n_j \times E}$ denote the matrix of embedding of sequence $\mathcal{S}^j$. We extract a

*context sequence embedding* $\mathbf{c}_j \in \mathbb{R}^E$ by applying a transformer encoder $\tilde{\mathbf{S}}^j = \Psi_{\text{enc}}^{\text{cont}}(\mathbf{S}^j) \in \mathbb{R}^{n_j \times E}$, followed by fixed-query attention

$$\mathbf{c}_j = \psi^{\text{cont}}(\mathbf{q}^{\text{cont}}, \tilde{\mathbf{S}}^j, \tilde{\mathbf{S}}^j) \in \mathbb{R}^E, \tag{8}$$

where $\mathbf{q}^{\text{cont}} \in \mathbb{R}^E$ is a learnable query.

We emphasize that we use an hierarchical approach where every sequence gets processed independently by $\Psi_{\text{enc}}^{\text{cont}}$ and encoded into a single embedding $\mathbf{c}_j$. This is much more scalable compared to combining all events into a single long sequence. The embeddings of all sequences $\mathbf{C} = [\mathbf{c}_1, \ldots, \mathbf{c}_m] \in \mathbb{R}^{m \times E}$ are finally combined by another transformer encoder $\tilde{\mathbf{C}} = \Psi_{\text{enc}}^{\text{comb}}(\mathbf{C}) \in \mathbb{R}^{m \times E}$.

**Context-aware History Encoding:** To encode the history $\mathcal{H}_t$ of events prior to time $t > t_{n_{\text{hist}}}^{\text{hist}}$, we embed each tuple $(t_i^{\text{hist}}, \kappa_i^{\text{hist}}, \Delta t_i^{\text{hist}})$ into feature vectors $\mathbf{H} = [\mathbf{u}_1^{\text{hist}}, \ldots, \mathbf{u}_{n_{\text{hist}}}^{\text{hist}}] \in \mathbb{R}^{n_{\text{hist}} \times E}$, reusing the networks from Eq. 7. These history embeddings serve as the queries of a transformer decoder $\Psi_{\text{dec}}^{\text{hist}}$, which attends to the context representation $\tilde{\mathbf{C}}$ (used as keys and values), yielding a unified encoding $\mathbf{h}_t^{\text{hist}}$ that integrates both history and context:

$$\mathbf{h}_t^{\text{hist}} = \Psi_{\text{dec}}^{\text{hist}}(\mathbf{H}, \tilde{\mathbf{C}}) \in \mathbb{R}^E. \tag{9}$$

**Intensity Parametrization:** To extract intensity functions for all marks from $\mathbf{h}_t^{\text{hist}}$, we concatenate $\mathbf{h}_t^{\text{hist}}$ with a (linear) encoding of $\kappa'$ to form $\mathbf{v}_{\kappa'}^{\mathcal{H}_t} = [\mathbf{h}_t^{\text{hist}}, \phi_\kappa^{\text{hist}}(\kappa')] \in \mathbb{R}^{2E}$, and project it to non-negative parameter estimates

$$\hat{\alpha}_{\kappa'}^{\mathcal{H}_t} = \Phi_\alpha(\mathbf{v}_{\kappa'}^{\mathcal{H}_t}) \in \mathbb{R}_+, \quad \hat{\beta}_{\kappa'}^{\mathcal{H}_t} = \Phi_\beta(\mathbf{v}_{\kappa'}^{\mathcal{H}_t}) \in \mathbb{R}_+ \quad \text{and} \quad \hat{\mu}_{\kappa'}^{\mathcal{H}_t} = \Phi_\mu(\mathbf{v}_{\kappa'}^{\mathcal{H}_t}) \in \mathbb{R}_+. \tag{10}$$

We enforce non-negativity via softplus output activations. These parameters define our neural conditional intensity estimate:[2]

$$\hat{\lambda}(t, \kappa' \mid \mathcal{H}_t) = \hat{\mu}_{\kappa'}^{\mathcal{H}_t} + (\hat{\alpha}_{\kappa'}^{\mathcal{H}_t} - \hat{\mu}_{\kappa'}^{\mathcal{H}_t}) \exp\left(-\hat{\beta}_{\kappa'}^{\mathcal{H}_t}(t - t_{n_{\text{hist}}})\right). \tag{11}$$

This parametrization is both flexible and interpretable. Indeed, immediately after incorporating a new event into the history, the intensity jumps to $\hat{\alpha}_{\kappa'}^{\mathcal{H}_t}$. Over long inter-event intervals, the intensity relaxes toward $\hat{\mu}_{\kappa'}^{\mathcal{H}_t}$. The relaxation rate is governed by $\hat{\beta}_{\kappa'}^{\mathcal{H}_t}$.

Although this parametrization may appear tailored to Hawkes-type dynamics, its parameters $\hat{\mu}_{\kappa'}^{\mathcal{H}_t}$, $\hat{\alpha}_{\kappa'}^{\mathcal{H}_t}$, and $\hat{\beta}_{\kappa'}^{\mathcal{H}_t}$ are themselves history- and mark-dependent. As a result, the model can represent rich local intensity behaviors, including localized triggering patterns (e.g., Rayleigh- or power-law-like kernels) and time-dependent baseline intensities. We highlight this empirically in Section 5.

**Training:** To train `FIM-PP` on a set of sequences $\mathcal{C}_\lambda$ from our synthetic pretraining data, we select a *target sequence* $\mathcal{T} \in \mathcal{C}_\lambda$ to provide a history of events and use remaining sequences $\mathcal{C}_\lambda \setminus \{\mathcal{T}\}$ as context. We subsample $\mathcal{C}_\lambda$, truncate sequences and vary the number of marks throughout training, which enables us to apply a pretrained `FIM-PP` in a wide range of (real-world) settings. Our train objective is the next-event negative log likelihood of the target sequence:

$$\mathcal{L}_{\text{NLL}} = \sum_{\kappa \in \mathcal{K}} \int_0^T \hat{\lambda}(s, \kappa \mid \mathcal{H}_s) ds - \sum_{(t, \kappa) \in \mathcal{T}} \hat{\lambda}(t, \kappa \mid \mathcal{H}_t). \tag{12}$$

We remark that we also experimented with a supervised sMAPE loss, the results of which can be found in the Appendix Tables, starting at Table 3. We found that this approach performs similarly well but is more computationally expensive and less flexible, which is why we use the NLL model in the main text. Appendix D discusses the training of `FIM-PP` in greater detail.

**Finetuning:** `FIM-PP` can be finetuned on the train split $\mathcal{C}$ of an evaluation dataset, minimizing $\mathcal{L}_{\text{NLL}}$. For each iteration, a random sequence $\mathcal{T} \in \mathcal{C}$ in the train split is selected as the target sequence. The remaining sequences $\mathcal{C} \setminus \{\mathcal{T}\}$ serve as context. Finetuning progress is monitored by processing target sequences from the validation split, given the train split context.

---

[2]Note that the functional form of $\hat{\lambda}$ is similar to the conditional intensity in Zhang et al. (2020).

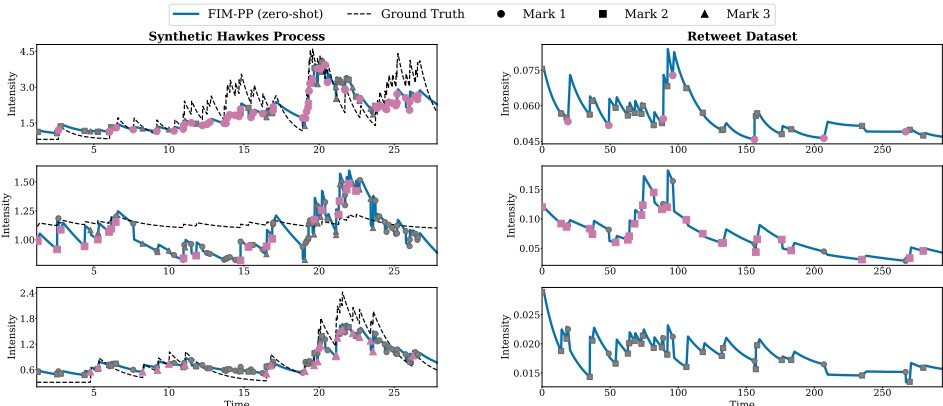

Figure 2: Example intensity estimates of `FIM-PP` on a synthetic Hawkes process with three marks, constant base intensity and exponential decaying kernels (left) and a real-world RETWEET dataset (right). Each row contains the intensity for one mark. Events of the same mark are colored magenta, while events for other marks are gray. For the Hawkes process, the model (blue line) estimate matches the ground-truth intensity level (black dashed line) closely. For the RETWEET data, `FIM-PP` estimates a mixture of many excitatory and a few inhibitory interactions.

## 5 EXPERIMENTS

In this section, we repeat the experiments by Zeng et al. (2024), who introduced `CDiff`, a recent state-of-the-art diffusion-based marked event sequence forecasting model. They compare their method against a range of intensity-based and intensity-free baselines, on common benchmark datasets, evaluated on standard metrics. Moreover, they made their code and exact evaluation dataset splits available[3], which allows us to directly compare against their results. In the following, we recall their experimental setup, describe the pretraining and application of `FIM-PP`, before presenting and analyzing our results.

### 5.1 EXPERIMENTAL SETUP

**Prediction Task:** Given a sequence of events $\mathcal{S} = \{(t_i, \kappa_i)\}_{i=1}^n$, the task is to predict the *next* $N \in \mathbb{N}$ *events* following $\mathcal{S}$, where $N$ is the *prediction horizon length*. We denote the ground-truth continuation by $\mathcal{S}_{\text{final}}^N$ and the model prediction by $\hat{\mathcal{S}}_{\text{final}}^N$. We refer to the case $N = 1$ as *next-event prediction* and to $N > 1$ as *multi-event prediction*.

**Evaluation Metrics:** We evaluate predictions by comparing $\hat{\mathcal{S}}_{\text{final}}^N$ with $\mathcal{S}_{\text{final}}^N$ across five metrics. For $N > 1$, we report the Optimal Transport Distance (OTD) (Mei et al., 2019); event count error ($\text{RMSE}_e$), comparing the number of predicted and true events per mark; and two standard regression metrics on waiting times: $\text{RMSE}_{\Delta t}$ and $\text{sMAPE}_{\Delta t}$. For the special case $N = 1$, OTD and $\text{RMSE}_e$ are not applicable, and we instead report next-event mark prediction accuracy (Acc). Formal definitions of all metrics are provided in Appendix F.

**Evaluation Data:** We benchmark on five widely used real-world datasets: TAXI, TAOBAO, STACK-OVERFLOW, AMAZON, and RETWEET. These datasets vary in the number of marks, sequence lengths, and event counts, making them a strong testbed for evaluating the broad applicability of `FIM-PP`. We use the preprocessing and train/test/validation splits of Zeng et al. (2024). Appendix E contains further details, including dataset statistics and original sources.

**Baselines:** We compare `FIM-PP` against methods falling into two categories: models that learn joint distributions over multiple events, and autoregressive approaches such as `FIM-PP`. The first category includes `Dual-TPP` (Deshpande et al., 2021), `HYPRO` (Xue et al., 2022), and the Cross-diffusion Model (`CDiff`) (Zeng et al., 2024). The second category further splits into intensity-based and intensity-free approaches. Intensity-based baselines are the Neural Hawkes Process (`NHP`) (Mei

---
[3]https://github.com/networkslab/cdiff

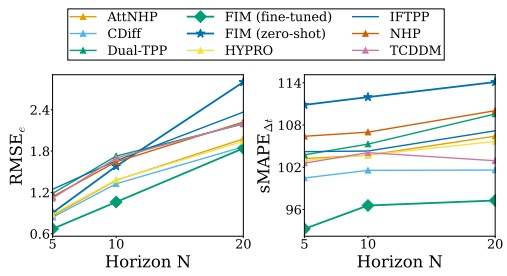

(a) Mean performance metrics across five different datasets for different horizon lengths.

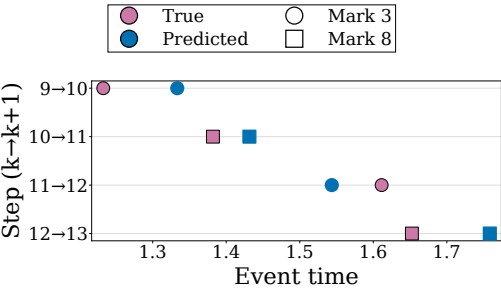

(b) Next event prediction on Taxi after fine-tuning

Figure 3: (a) shows that `FIM-PP` (zs) is competitive but slightly worse than the baseline models. `FIM-PP` (f) however performs best among all horizon lengths. (b) shows that `FIM-PP` (f) also reliably captures patterns in the Taxi dataset.

& Eisner, 2017), and the Attentive Neural Hawkes Process (`A-NHP`) (Mei et al., 2022). Intensity-free baselines are the Intensity-Free Temporal Point Process (`IFTPP`) (Shchur et al., 2020), and the Temporal Conditional Diffusion Denoising Model (`TCDDM`) (Lin et al., 2022).

## 5.2 PRETRAINING, FINE-TUNING AND INTENSITY PREDICTIONS

**Pretraining**. We pretrain a single `FIM-PP` on a synthetic dataset containing 14.4M events, simulated from 72K point processes of diverse kernels, and sparsity levels, and varying number of marks, sequences, and events. Appendix B contains the details. The model has 16M parameters and supports up to $|\mathcal{K}| = 22$ marks to cover all evaluation datasets. Further pretraining details are provided in Appendix D.

**Finetuning**. In *zero-shot mode*, we apply the pretrained model directly to all evaluation datasets, and label the results by `FIM-PP` (zs). We also experiment with *finetuning* `FIM-PP` on the train split of all evaluation datasets, and label these results by `FIM-PP` (f). `FIM-PP` utilizes up to 2000 sequences from the train split of an evaluation dataset as context, limited only by the maximum number of sequences seen during training. The ablations in Figure 6 also indicate that typically much less than 2000 context paths would be sufficient. The sequences in the test split are given as context to `FIM-PP`. Finally, for multi-event prediction, `FIM-PP` simulates events autoregressively, similar to other (intensity-based) baselines.

Due to the relatively small parameter size of `FIM-PP` and the pre-trained prior, fine-tuning can be performed quickly. For all datasets, fine-tuning was achieved in a few minutes, requiring not more than 11 GB of memory. This means that fine-tuning `FIM-PP` does not take longer than training the baseline models from scratch, which reportedly takes up to 4 hours (Xue et al., 2024, Sec. D.1). Fine-tuning is therefore feasible for users with small computational resources. Further speed-ups might be possible with LoRA (Hu et al., 2022).

**Intensity Predictions**. Figure 2 shows intensities inferred by `FIM-PP` in zero-shot mode, both from an unseen synthetic Hawkes process (in-distribution generalization), and from the RETWEET dataset (out-of-distribution generalization). Moreover, Figure 4 in the Appendix illustrates the performance of `FIM-PP` (zs) on various synthetic datasets and emphasizes that `FIM-PP` (zs) also generalizes well for processes with powerlaw kernels, a kernel type that was not present in the pretraining dataset. This confirms that our parameterization is general enough. In what follows, we quantitatively evaluate `FIM-PP`.

## 5.3 MULTI-EVENT PREDICTION

Table 1 reports OTD and sMAPE$_{\Delta t}$ results for $N = 20$ and four datasets. Remarkably, `FIM-PP` achieves competitive performance in *zero-shot mode*, matching or surpassing specialized baselines on TAXI and RETWEET data. This shows that, solely from pretraining on synthetic data, the model

Table 1: Performance on four real-world datasets, predicting $N = 20$ events. Results for baseline methods were extracted from Zeng et al. (2024). We report mean and standard deviation over 10 trials for two metrics. Best results are bold; second-best results are bold gray.

| Method | TAXI | | STACKOVERFLOW | | AMAZON | | RETWEET | |
| --- | --- | --- | --- | --- | --- | --- | --- | --- |
| | OTD | sMAPE$_{\Delta t}$ | OTD | sMAPE$_{\Delta t}$ | OTD | sMAPE$_{\Delta t}$ | OTD | sMAPE$_{\Delta t}$ |
| HYPRO | $21.60_{\pm0.20}$ | $93.8_{\pm0.4}$ | $42.40_{\pm0.20}$ | $111.00_{\pm0.60}$ | $38.6_{\pm0.5}$ | $82.5_{\pm0.8}$ | $61.03_{\pm0.09}$ | $106.11_{\pm1.51}$ |
| Dual-TPP | $24.48_{\pm0.38}$ | $95.2_{\pm0.2}$ | $41.75_{\pm0.20}$ | $117.58_{\pm0.42}$ | $42.6_{\pm0.7}$ | $86.5_{\pm2.0}$ | $61.10_{\pm0.10}$ | $106.90_{\pm1.29}$ |
| A-NHP | $24.76_{\pm0.22}$ | $97.4_{\pm0.4}$ | $42.59_{\pm0.41}$ | $108.54_{\pm0.53}$ | $39.5_{\pm0.3}$ | $84.3_{\pm1.8}$ | $60.63_{\pm0.10}$ | $107.23_{\pm1.29}$ |
| NHP | $25.11_{\pm0.27}$ | $96.5_{\pm0.5}$ | $43.79_{\pm0.15}$ | $116.95_{\pm0.40}$ | $42.6_{\pm0.3}$ | $92.1_{\pm1.6}$ | $60.95_{\pm0.08}$ | $107.08_{\pm1.40}$ |
| IFTPP | $24.05_{\pm0.61}$ | $95.7_{\pm0.8}$ | $46.28_{\pm0.89}$ | $115.12_{\pm0.63}$ | $43.8_{\pm0.2}$ | $90.9_{\pm1.6}$ | $61.72_{\pm0.15}$ | $106.71_{\pm1.62}$ |
| TCDDM | $22.15_{\pm0.53}$ | $90.6_{\pm0.6}$ | $42.13_{\pm0.59}$ | $107.66_{\pm0.93}$ | $42.2_{\pm0.2}$ | $83.8_{\pm1.5}$ | $60.50_{\pm0.09}$ | $106.05_{\pm0.61}$ |
| CDiff | $\mathbf{21.01}_{\pm0.16}$ | $\mathbf{88.0}_{\pm0.2}$ | $\mathbf{41.25}_{\pm1.40}$ | $106.18_{\pm0.34}$ | $\mathbf{37.7}_{\pm0.2}$ | $\mathbf{82.0}_{\pm1.9}$ | $60.66_{\pm0.10}$ | $106.18_{\pm1.12}$ |
| FIM-PP (zs) | $23.15_{\pm0.07}$ | $\mathbf{76.8}_{\pm0.4}$ | $49.26_{\pm0.06}$ | $\mathbf{96.36}_{\pm0.05}$ | $46.2_{\pm0.1}$ | $128.6_{\pm0.4}$ | $\mathbf{60.24}_{\pm0.16}$ | $\mathbf{99.07}_{\pm0.39}$ |
| FIM-PP (f) | $\mathbf{17.91}_{\pm0.12}$ | $\mathbf{76.8}_{\pm0.5}$ | $\mathbf{39.80}_{\pm0.04}$ | $\mathbf{88.25}_{\pm0.19}$ | $\mathbf{37.2}_{\pm0.1}$ | $\mathbf{81.2}_{\pm0.1}$ | $\mathbf{59.44}_{\pm0.08}$ | $\mathbf{87.59}_{\pm0.17}$ |

can translate contextual patterns into accurate multi-event predictions, without any further training or supervision.

The same table also demonstrates the effectiveness of finetuning. The finetuned FIM-PP (f) consistently outperforms both FIM-PP (zs) and all baselines, across the four datasets and nearly all metrics. Additional experiments with shorter horizons ($N = 10, 5$) and alternative metrics ($\text{RMSE}_e$, $\text{RMSE}_{\Delta t}$) are reported in Appendix A, providing a complementary view.

Figure 3a summarizes performance across horizon lengths by averaging results over all datasets. In aggregate, FIM-PP (zs) performs on par with the baselines, whereas FIM-PP (f) consistently outperforms them, in agreement with our previous analysis.

We attribute the effectiveness of finetuning to two factors: (i) the strong prior encoded in the model weights through pretraining on our synthetic distribution, which provides a favorable initialization for finetuning; and (ii) the flexibility of the foundation model architecture, which enables *direct access* to patterns in the training split *at evaluation time*, since these patterns can be retrieved from the context. The first point is supported by Figure 5 in the Appendix, which shows that the pretrained model converges faster and achieves better final performance than the same architecture trained from scratch.

## 5.4 NEXT-EVENT PREDICTION

The *next-event prediction* task is a special case of multi-event prediction, but it differs in nature. Whereas multi-event prediction evaluates the *distributional quality* of a set of future events, next-event prediction requires accurately forecasting *one specific* event time and mark. This distinction is reflected in the evaluation metrics (see Appendix F), and is particularly important for mark prediction: small systematic errors in the local ordering of marks can strongly affect next-event accuracy, even if the model captures the longer-horizon event statistics well.

Table 2 reports next-event prediction results ($N = 1$) on two real-world datasets. On TAXI, FIM-PP (zs) performs competitively on event-time prediction, but struggles with mark accuracy. This behavior is consistent with the structure of the dataset: many TAXI sequences exhibit a near-deterministic alternation between two marks (see Fig. 9 and Appendix G). Such transition-like patterns are not explicitly represented in our synthetic point-process prior. As a result, FIM-PP (zs) can infer reasonable event times from the context, but does not reliably recover the precise alternating mark rule in zero-shot mode.

The picture is different for TAOBAO. Here, FIM-PP (zs) outperforms all baselines on event-time prediction and also achieves the best zero-shot mark accuracy. A natural explanation is that the dominant structure of TAOBAO is better aligned with our pretraining distribution (see Fig. 10 in the Appendix). Indeed, a dataset dominated by one mark can be represented by strongly imbalanced mark-specific base intensities, while long waiting times are compatible with low-rate Poisson-like

Table 2: Next-event prediction performance on two real-world datasets, displayed with mean and standard deviation over 10 trials. Results for baseline methods were extracted from Zeng et al. (2024) (we remark that they did not report next-event prediction results for the other datasets). Best results are bold; second-best results are bold gray.

| | **TAXI** | | | **TAOBAO** | | |
|---|---|---|---|---|---|---|
| Method | $\mathrm{RMSE}_{\Delta t}$ | Acc | $\mathrm{sMAPE}_{\Delta t}$ | $\mathrm{RMSE}_{\Delta t}$ | Acc | $\mathrm{sMAPE}_{\Delta t}$ |
| A-NHP | **0.32** $\pm 0.00$ | **0.91** $\pm 0.01$ | **85.13** $\pm 0.26$ | 0.53 $\pm 0.00$ | 0.47 $\pm 0.01$ | 129.13 $\pm 1.35$ |
| Dual-TPP | 0.34 $\pm 0.01$ | **0.91** $\pm 0.01$ | 89.12 $\pm 0.75$ | 0.53 $\pm 0.01$ | 0.47 $\pm 0.02$ | 131.43 $\pm 1.99$ |
| NHP | 0.34 $\pm 0.01$ | **0.91** $\pm 0.01$ | 90.63 $\pm 0.61$ | 0.53 $\pm 0.00$ | 0.46 $\pm 0.01$ | 133.69 $\pm 2.25$ |
| IFTPP | 0.38 $\pm 0.01$ | 0.90 $\pm 0.01$ | 90.03 $\pm 0.47$ | 0.53 $\pm 0.01$ | 0.45 $\pm 0.01$ | 126.01 $\pm 1.48$ |
| CDiff | 0.34 $\pm 0.01$ | **0.91** $\pm 0.00$ | 87.12 $\pm 0.61$ | 0.52 $\pm 0.01$ | 0.48 $\pm 0.00$ | 127.12 $\pm 1.36$ |
| FIM-PP (zs) | 0.34 $\pm 0.01$ | 0.47 $\pm 0.03$ | 98.03 $\pm 0.92$ | **0.13** $\pm 0.00$ | 0.52 $\pm 0.01$ | **112.89** $\pm 0.66$ |
| FIM-PP (f) | 0.45 $\pm 0.03$ | **0.91** $\pm 0.00$ | 85.91 $\pm 0.86$ | 0.16 $\pm 0.00$ | 0.60 $\pm 0.01$ | 115.98 $\pm 0.72$ |

We remark that the released source code of CDIFF Zeng et al. (2024) only predicts the last event per sequence for next event prediction. We believe that this is a bug because it also produced very unreasonable results. We hence report the performance for all events, not just the last.

regimes, weakly interacting Hawkes processes, and inhibitory interactions. These patterns are naturally covered by our prior.

Finetuning further improves adaptation to dataset-specific structure. On TAXI, FIM-PP (f) recovers the alternating pattern much more reliably, as illustrated in Figure 3, and its mark accuracy rises substantially. On TAOBAO, finetuning further improves mark accuracy, suggesting that the pretrained model already provides a useful prior and that target-specific optimization mainly sharpens the mark distribution. Interestingly, the zero-shot model remains slightly better on event-time errors for TAOBAO, indicating a mild trade-off between adapting the mark distribution and preserving the pretrained timing prior.

Overall, these results suggest that the current pretraining distribution is already well suited for datasets governed by mark imbalance and heterogeneous waiting times, but less suited for highly regular mark-transition patterns. In Appendix G, we therefore suggest broadening the synthetic pretraining distribution with process families that explicitly include alternating, cyclic, or transition-matrix-like mark dynamics.

## 6 CONCLUSIONS

In this work, we introduced FIM-PP, the first *Foundation Inference Model* for inferring marked temporal point processes (MTPPs) from real-world data. Our experiments show that a *single* FIM-PP, pretrained only on synthetic MTPP data, can already match the predictive performance of existing intensity-based MTPP methods in *zero-shot mode*, i.e., *without any further training*. *Finetuning* on target data further improves results within only a few iterations, enabling FIM-PP to *outperform competing methods on the majority of evaluated tasks*.

**Limitations:** Although our pretraining distribution is broad, it is not universal and therefore cannot capture many real-world patterns. As a result, zero-shot performance may degrade under distribution shift. FIM-PP is further constrained by the maximum number of marks $|\mathcal{K}|$ and the maximum sequence length used during training. When these limits are exceeded, the model may not fully exploit the available context.

**Future Work:** An important direction for future work is to *broaden the pretraining distribution* beyond the parametrization in Eq. 6, with the goal of capturing a wider range of data patterns in zero-shot mode, and providing an even stronger initialization for finetuning. In addition, *intensity-free* methods have recently shown strong predictive performance (Panos, 2024). We plan to investigate how such methods can be incorporated into our amortized in-context learning framework (first results can be found in Berghaus (2026)).

## 7 REPRODUCIBILITY STATEMENT

Our core methodology consists of two parts: *synthetically generated training data* and a *foundation inference model* for marked temporal point process. *Data generation* is described extensively in Section 4.1 and complemented by Appendix B, which covers the exact hyperparameters and design choices required to reproduce our training dataset. Section 4.2 describes the architecture of `FIM-PP`. *Training details*, including hyperparameter choices and submodule sizes, are described in Appendix D. Our pretrained model, repository, and tutorials are available online[4] .

The *real-world datasets* used in our experiments are described in Appendix E, including dataset sizes and numbers of marks. For data sourcing and preprocessing, we follow Zeng et al. (2024), as discussed in Appendix E. Finally, the *evaluation metrics* used in all experiments are described in Appendix F.

## ACKNOWLEDGMENTS

This research has been funded by the Federal Ministry of Education and Research of Germany and the state of North-Rhine Westphalia as part of the Lamarr Institute for Machine Learning and Artificial Intelligence. Additionally, César Ojeda was supported by Deutsche Forschungsgemeinschaft (DFG) – Project-ID 318763901 – SFB1294.

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

# A ADDITIONAL RESULTS

## A.1 PERFORMANCE ON VARIOUS SYNTHETIC DATASETS

In this section we highlight the performance of `FIM-PP` (zs) on various synthetic datasets coming from different processes. Figure 4 compares the estimated intensity to the ground-truth on some synthetic processes, including the powerlaw kernel. The model captures the essence of all processes, including the out-of-distribution powerlaw kernel.

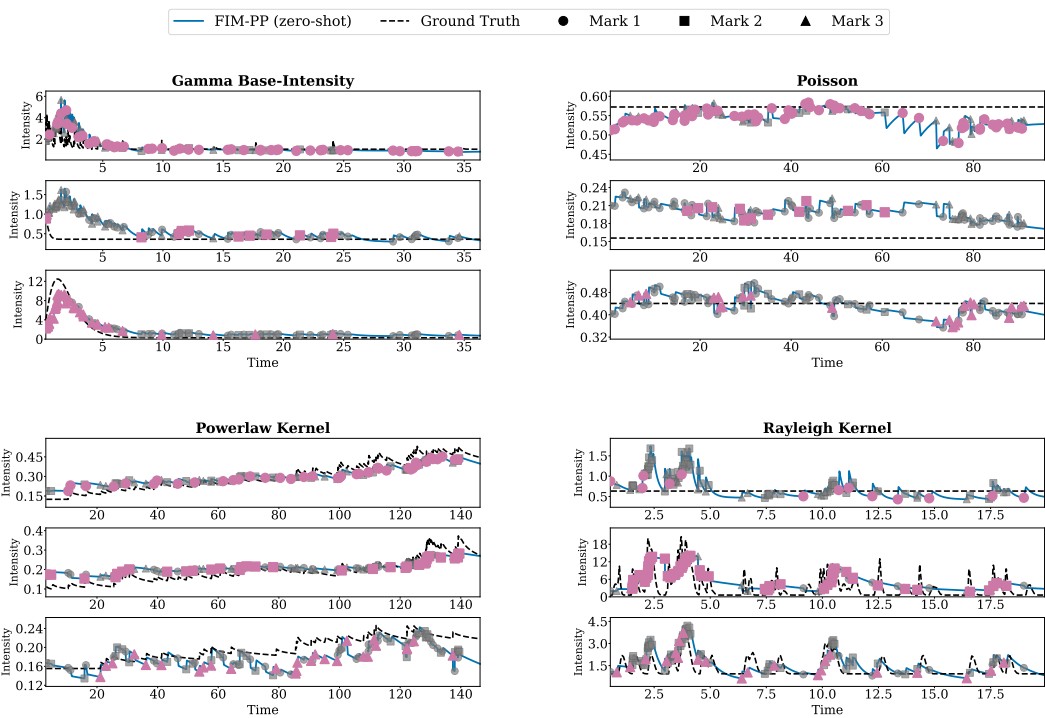

Figure 4: Intensity predictions of `FIM-PP` (zs) on synthetic datasets of four different process types. We remark that the model has not been trained on powerlaw kernels but still predicts them with decent accuracy.

## A.2 LONG-HORIZON PREDICTION

The experimental setup defined by Zeng et al. (2024) covers four metrics (OTD, $\text{RMSE}_e$, $\text{RMSE}_{\Delta t}$, $\text{sMAPE}_{\Delta t}$) and five real-world datasets (TAXI, TAOBAO, STACKOVERFLOW, AMAZON and RETWEET). Table 3, Table 4 and Table 5 contain the long horizon results for all these datasets and metrics for horizon sizes $N = 20$, $N = 10$ and $N = 5$, respectively.

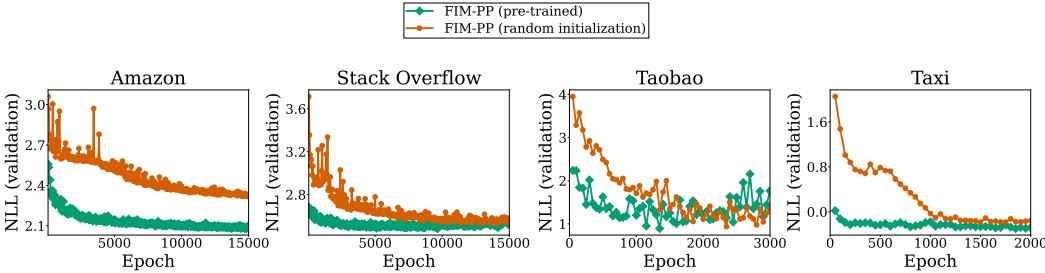

Figure 5: Comparison of the fine-tuning loss curves of a pre-trained `FIM-PP` model versus random initialization. Note that one epoch corresponds to just one inference-path prediction and is therefore very fast. Our results indicate that the pre-training achieves faster convergence as well as a higher loglikelihood when converged.

## A.3 FINE-TUNING

Due to the relatively small parameter count of 16.1M, finetuning takes just a few minutes and less than 11GB of GPU memory for all datasets. In Figure 5 we compare the finetuning speed against training a `FIM-PP` from scratch, i.e. with randomly initialized parameters. The results indicate that pre-training yields a good initialization for finetuning, which converges much faster than training from randomly initialized set of parameters. Moreover, the finetuned model generally archives lower errors on the test split.

## A.4 PERFORMANCE WITH VARYING CONTEXT SIZE

We also investigated the sensitivity of `FIM-PP` to the number of context paths passed to the model. We studied this behavior on data from three synthetic processes and evaluated the results based on the sMAPE metric. Figure 6 contains the results of this experiment. Initially, providing more context paths to the model improves the accuracy of the estimated process. After a few hundred context paths, the performance saturates; more paths only improve marginally improve the estimate. Crucially, the performance does not degrade with more paths, highlighting the robustness of our model.

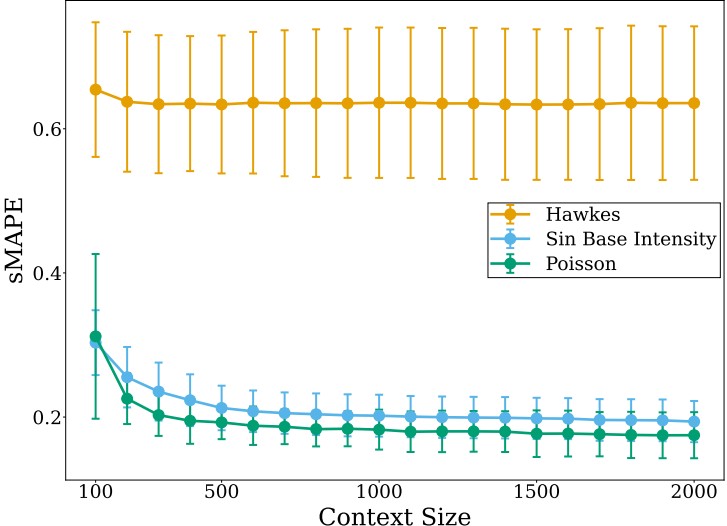

Figure 6: sMAPE error against the ground truth intensity for varying number of context paths. We used 100 points per path. Our results indicate that the performance of the model does not noticeable improve for more than 500 paths, at least for the synthetic datasets tested.

## B   DATA GENERATION

To train our Foundation Inference Model, we generate a comprehensive synthetic dataset of marked temporal point processes. Each process is an instance of a multivariate Hawkes process. The conditional intensity function $\lambda(t, \kappa \mid \mathcal{H}_t)$ at time $t \in \mathbb{R}_+$ for a mark $\kappa \in \mathcal{K}$ given a history of marked events $\mathcal{H}_t = \{(t_i, \kappa_i) \mid t_i < t\}$ is defined as

$$\lambda(t, \kappa \mid \mathcal{H}_t) = \max\left(0, \mu_\kappa(t) + \sum_{(t', \kappa') \in \mathcal{H}_t} z_{\kappa\kappa'} \gamma_{\kappa\kappa'}(t - t')\right), \tag{13}$$

where $\mu_\kappa$ is the time-dependent base intensity for mark $\kappa$, $\gamma_{\kappa\kappa'}$ is the interaction kernel between $\kappa$ and $\kappa' \in \mathcal{K}$ and $z_{\kappa\kappa'} \in \mathbb{R}$ are sampled pre-factors of the interaction, varying the interaction behavior further.

To the best of our knowledge, no open-source solution for sampling such Hawkes processes with time-dependent base intensity functions exists. Hence, we implemented an efficient custom sampling library for such processes in C++. We will release the source code of this library in the supplementary material of our work.

### B.1   DATASET CONFIGURATIONS

We sample Hawkes processe instances over a set of marks $\mathcal{K}$ in equation 13 in two stages.

At first, the functional forms for the base intensities $\mu_\kappa$ and interaction kernels $\gamma_{\kappa\kappa'}$ are drawn from a library of parametric functions. The parameters for these functions are then sampled from specified prior distributions. Our used functional forms and their parameters are summarized in Table 6. The parameter ranges were chosen more or less arbitrarily so that the paths look *realistic*. We keep these choices fixed and did not modify them based on the performance on our evaluation sets in order to prevent overfitting to those, which would be contrary to the concept of a foundation model.

The pre-factors further diversify the sampled processes by introducing sparse connectivity and inhibitory effects. For each process with interactions, we choose one of two pre-factor distributions $Z_{\text{strong}}$ and $Z_{\text{sparse}}$ on $\{-1, 0, 1\}$, which differ by their induced connectivity:

$$Z_{\text{strong}} = \text{Categorical}\left(-1 : 0.06,\ 0 : 0.4,\ 1 : 0.54\right) \tag{14}$$
$$Z_{\text{sparse}} = \text{Categorical}\left(-1 : 0.01,\ 0 : 0.9,\ 1 : 0.09\right) \tag{15}$$

In other words, for $Z_{\text{strong}}$, only $40\%$ of interactions will be non-influencing, while for $Z_{\text{sparse}}$, $90\%$ of interactions will be non-influencing. For influencing interactions, $90\%$ will be excitatory, while $10\%$ will be inhibitory.

Once the full intensity function for a process is defined, event sequences are generated using Ogata's modified thinning algorithm.

Figure 7 contains summarizing statistics of our train distribution. Aggregated, this prior is very broad. Importantly, it covers distributions of the real-world datasets in our experiments. The corresponding statistics for these datasets are depicted in Figures 8 to 12.

### B.2   DATASET SIZE

We sample instances of every Hawkes process configuration in Table 6, and simulate them for different number of marks, sequences and events, detailed in Table 7. In total, our training data consisted of 72k processes and 14.4M events.

## C   INSTANCE NORMALIZATION

To ensure that `FIM-PP` can generalize across datasets with vastly different time scales, we introduce an instance normalization scheme that makes the model agnostic to the absolute units of time. Let $\mathcal{C} = \{\mathcal{S}^j\}_{j=1}^m$ denote a context of `FIM-PP`, i.e. a set of marked event sequences $\mathcal{S}^j = \{(t_i^j, \kappa_i^j)\}_{i=1}^{n_j}$.

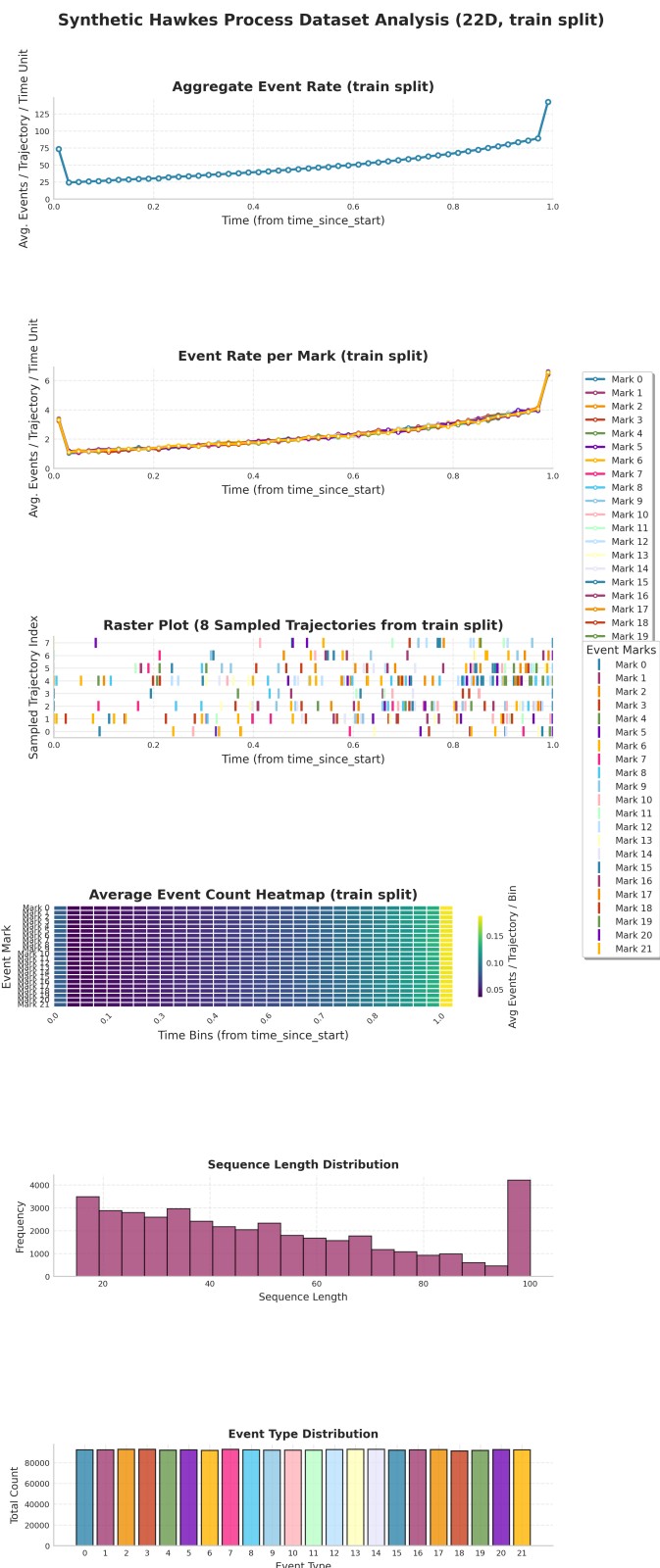

Figure 7: Distribution of our synthetic training data for Hawkes processes with 22 marks. The distribution is almost uniform and very broad and therefore also captures real world phenomena.

Figure 8: Amazon dataset statistics.

Figure 9: Taxi dataset statistics.

Figure 10: Taobao dataset statistics.

Figure 11: Stackoverflow dataset statistics.

Figure 12: Retweet dataset statistics.

Identifying $t_0^j = 0$, we define the inter-event times as $\Delta t_i^j = t_i^j - t_{i-1}^j$ and the maximum inter-event time in the context as

$$\Delta t_{\max}^{\text{cont}} = \max_{j=1,\ldots,m} \max_{i=1,\ldots,n_j} \Delta t_i^j \quad . \tag{16}$$

All time-related inputs to the model, including context event times $t_i^j$, inter-event times features $\Delta t_i^j$, and history event times (e.g. from a target sequence during training) $t$, are scaled by the maximum inter-event time:

$$t' = \frac{t}{\Delta t_{\max}^{\text{cont}}}. \tag{17}$$

This transformation maps all temporal information to a canonical scale where the largest inter-event gap becomes $1$.

This change of time variable also transforms the intensity function. To preserve the number of expected events within a differential interval, the intensities must be related by $\lambda(t)dt = \lambda'(t')dt'$. Since $dt = \Delta t_{\max}^{\text{cont}} dt'$, it follows that the intensity in the normalized time domain, $\lambda'(t')$, is a scaled version of the original:

$$\lambda'(t') = \Delta t_{\max}^{\text{cont}} \cdot \lambda(t). \tag{18}$$

Consequently, the model is trained to predict this normalized intensity $\lambda'(t')$. During inference, to obtain the intensity in the original, real-world time scale, the model's output is simply denormalized by dividing by the same constant $\Delta t_{\max}^{\text{cont}}$. This entire process allows the FIM to learn scale-invariant temporal dynamics, a key requirement for effective zero-shot inference on unseen data.

## D  TRAINING DETAILS

Our Foundation Inference Model was trained on the comprehensive synthetic datasets described in Appendix B. The training took about 5 days on a single NVIDIA A100-80GB.

### D.1  DATA HANDLING AND BATCHING

Each sample in our dataset represents a single underlying process, comprising up to 2000 distinct time series paths. During training, we dynamically partition these paths into context and inference sets for each batch.

**On-the-fly Path Selection**   For each sample, we randomly select a single path ($P_{\text{inference}} = 1$) to serve as the inference target. The remaining paths are designated as the context set. To train a model that is robust to varying amounts of contextual information, the number of context paths presented in each training step is randomized. Specifically, for each sample in a batch, we uniformly sample a number of context paths between a minimum of 400 and a maximum of 2000.

**Variable Sequence Lengths**   As a form of data augmentation, we also vary the length of the historical sequences. For 90% of the training batches, all sequences (both context and inference) are truncated to a random length chosen uniformly from the interval $[15, 100]$. For the remaining 10% of batches, the full sequence length of 100 events is used. This strategy encourages the model to make reliable predictions from both short and long historical contexts. For validation, we use fixed, full-length sequences to ensure consistent and comparable evaluation metrics.

### D.2  HYPERPARAMETERS AND OPTIMIZATION

The model architecture is based on the Transformer Vaswani et al. (2017). Context sequences are processed by a 4-layer Transformer encoder, and the resulting path summaries are further refined by a 2-layer Transformer encoder. The history of the target sequence is processed by a 4-layer Transformer decoder, which attends to the context summary as memory. Both encoders and the decoder use 4 attention heads and a hidden dimension of 256. The final intensity parameters $(\mu, \alpha, \beta)$ are predicted by three separate Multi-Layer Perceptrons (MLPs), each with two hidden layers of 256 units.

In total, our model has 16.1 million trainable parameters.

We trained the model using the AdamW optimizer Loshchilov & Hutter (2019) with a learning rate of $5 \times 10^{-5}$ and a weight decay of $10^{-4}$. To accelerate computation, we utilized bfloat16 mixed-precision training.

### D.3 TRAIN OBJECTIVE

We use the standard negative log-likelihood (NLL) for a marked temporal point process as the train objective for FIM-PP. By Section 3, the MTPP density at a sequence of events $\mathcal{S} = \{(t_i, \kappa_i)\}_{i=1}^n$ in the interval $[0, T]$ is

$$f\left(\{(t_i, \kappa_i)\}_{i=1}^n\right) = \left[\prod_{i=1}^n \lambda(t_i, \kappa_i \mid \mathcal{H}_{t_i})\right] \exp\left(-\int_0^T \lambda(s \mid \mathcal{H}_s)ds\right) . \tag{19}$$

Thus, the NLL of a target sequence under the distribution induced the model's predicted intensity function $\hat{\lambda}$ is

$$\mathcal{L}_{\text{NLL}} = \sum_{\kappa \in \mathcal{K}} \hat{\Lambda}(T, \kappa) - \sum_{(t, \kappa) \in \mathcal{T}} \hat{\lambda}(t, \kappa \mid \mathcal{H}_t) . \tag{20}$$

where $\hat{\Lambda}(T, \kappa) = \int_0^T \hat{\lambda}(s, \kappa \mid \mathcal{H}_s)ds$ is the predicted integrated intensity. We approximate the integral using Monte Carlo integration

$$\hat{\Lambda}(T, \kappa) \approx \frac{T}{N_{\text{MC}}} \sum_{i=1}^{N_{\text{MC}}} \hat{\lambda}(s_i, \kappa \mid \mathcal{H}_{s_i}), \tag{21}$$

with $N_{\text{MC}} = 100$ samples and $s_i \sim \mathcal{U}(0, T)$.

## E EVALUATION DATASETS

To evaluate the inference capabilities of FIM-PP, we use five widely-recognized real-world datasets that were not seen during training:

**AMAZON**   This dataset comprises sequences of product reviews from users on the Amazon platform, collected over a ten-year period from 2008 to 2018 Ni et al. (2019). Each sequence represents the review history of a single user. An event is defined by the timestamp of a review, and its mark corresponds to one of 16 distinct product categories. The analysis is performed on a subset of 5,200 of the most active users to ensure sequences are sufficiently long for meaningful analysis.

**TAXI**   Derived from New York City's public taxi trip records, this dataset captures the operational patterns of taxi drivers. Each sequence corresponds to the activity log of an individual driver. Events are either pick-ups or drop-offs, and the event marks are defined by the combination of the event type (pick-up/drop-off) and the borough where it occurred, resulting in 10 unique marks. The dataset consists of sequences from a random sample of 2,000 drivers.

**TAOBAO**   This dataset originates from the 2018 Tianchi Big Data Competition and contains logs of user interactions on the Taobao e-commerce platform over a period in late 2017 Zhu et al. (2018). The sequences track the behavior of anonymous users, including actions like browsing and purchasing. The 17 event types correspond to different product category groups. For the evaluation, sequences from the 2,000 most active users are utilized.

**STACKOVERFLOW**   Sourced from the popular question-and-answering website StackOverflow, this dataset tracks the awarding of achievement badges to users over a two-year span Leskovec & Krevl (2014). Each sequence represents a user's history of earned badges. The events are the timestamps when badges were awarded, and the marks are the 22 different types of badges available on the platform. The evaluation subset includes 2,200 active users.

**RETWEET** This dataset tracks the dynamics of information spread through time-stamped user retweet sequences Zhou et al. (2013). Each sequence corresponds to the retweet history of an individual user. An event is defined by the timestamp of a retweet, and its mark is categorized into one of three types based on the influence of the original poster: "small" (fewer than 120 followers), "medium" (fewer than 1,363 followers), and "large" (all other users). The analysis is performed on a subset of 5,200 active users.

For all real-world datasets, we use the pre-processing and splits from Zeng et al. (2024).

To compare against the other models, `FIM-PP` uses the sequences which the other models used for training as context and used the same inference sequences for evaluation.

## F    EVALUATION METRICS

Following Zeng et al. (2024), we adopt a comprehensive set of metrics to evaluate both the temporal and categorical aspects of the predicted sequences. Let $\mathcal{S}_{\text{future}} = \{(t_i, \kappa_i)\}_{i=1}^{N}$ be a ground truth sequence of $N$ future events, and let $\hat{\mathcal{S}}_{\text{future}}$ be the corresponding predicted sequence. The metrics are defined based on the sequence of inter-arrival times $\boldsymbol{\Delta t} = [\Delta t_1, \ldots, \Delta t_N]$ (where $\Delta t_i = t_i - t_{i-1}$) and the sequence of marks.

**Optimal Transport Distance (OTD)** We use the Optimal Transport Distance (OTD) to provide a holistic measure of similarity between the predicted and ground truth event sequences (Mei et al., 2019). OTD calculates the minimum cost required to transform the predicted sequence $\hat{\mathcal{S}}_{\text{future}}$ into the ground truth sequence $\mathcal{S}_{\text{future}}$ through a series of operations (insertions, deletions, and substitutions), each associated with a cost. This metric effectively captures discrepancies in timing, marks, and the total number of events.

**RMSE on Event Counts (RMSE$_e$)** This metric evaluates how well the model captures the distribution of event types in the predicted sequence. For each event type $\kappa \in \mathcal{K}$, we count its occurrences in the ground truth sequence ($C_\kappa$) and the predicted sequence ($\hat{C}_\kappa$). The RMSE$_e$ is the root mean squared error over the vector of these counts, averaged across all $m$ test sequences:

$$\text{RMSE}_e = \sqrt{\frac{1}{m} \sum_{j=1}^{m} \sum_{\kappa \in \mathcal{K}} (C_{j,\kappa} - \hat{C}_{j,\kappa})^2} \tag{22}$$

**Event Type Accuracy (Acc)** This metric directly measures the model's ability to predict the correct event type at each position in the sequence. It is calculated as the fraction of events for which the predicted mark $\hat{\kappa}_i$ matches the ground truth mark $\kappa_i$, averaged over all test sequences. This provides a strict, position-wise evaluation of the categorical predictions.

$$\text{Acc} = \frac{1}{m} \sum_{j=1}^{m} \frac{1}{N} \sum_{i=1}^{N} \mathbb{I}(\kappa_{j,i} = \hat{\kappa}_{j,i}) \tag{23}$$

where $\mathbb{I}(\cdot)$ is the indicator function. Unlike RMSE$_e$, which assesses the overall distribution of event types, accuracy penalizes mispredictions at specific positions, making it a more challenging metric for sequential order. A higher accuracy indicates better performance.

**Time-series Forecasting Metrics** To specifically assess the accuracy of the predicted inter-arrival times $\boldsymbol{\Delta t}$, we report two standard time-series forecasting metrics.

- **RMSE on Inter-arrival Times (RMSE$_{\Delta t}$)**: The standard root mean squared error between the predicted and true vectors of inter-arrival times.

$$\text{RMSE}_{\Delta t} = \sqrt{\frac{1}{m} \sum_{j=1}^{m} \frac{1}{N} \sum_{i=1}^{N} (\Delta t_{j,i} - \hat{\Delta t}_{j,i})^2} \tag{24}$$

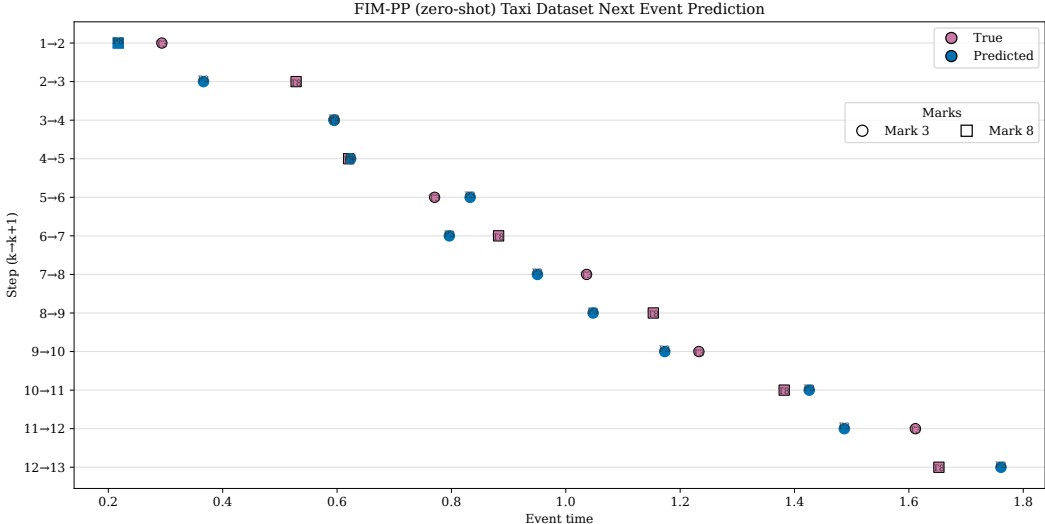

Figure 13: `FIM-PP` in zero-shot mode struggles to predict the next event type right if the dataset has alternating patterns such as here for the Taxi dataset.

- **Symmetric Mean Absolute Percentage Error (sMAPE$_{\Delta t}$):** A normalized version of MAPE that is less sensitive to outliers and zero values.

$$\text{sMAPE}_{\Delta t} = \frac{100}{m} \sum_{j=1}^{m} \frac{1}{N} \sum_{i=1}^{N} \frac{2|\Delta t_{j,i} - \hat{\Delta} t_{j,i}|}{|\Delta t_{j,i}| + |\hat{\Delta} t_{j,i}|} \tag{25}$$

## G  CHALLENGES IN NEXT EVENT PREDICTION

Our evaluations in table 2 reveal that `FIM-PP` in zero-shot mode already performs well for next event time prediction. It however gets a noticeably worse error in the next event type prediction. Up on investigating this, we found that many of the real-world datasets have specific patterns (such as oscillations between two marks) that `FIM-PP` (zs) struggles to capture (see fig 13). After fine-tuning, it is however able to spot these patterns well (see fig 14). This might also explain why `FIM-PP` performs better on long-horizon tasks: The specific order of the events does not matter here.

We hypothesize that the underlying reason for this shortcoming is that our synthetic dataset distributions does not capture these patterns well. We are planning to investigate this further and to update our synthetic distribution to include such patterns and provide an updated version of `FIM-PP`.

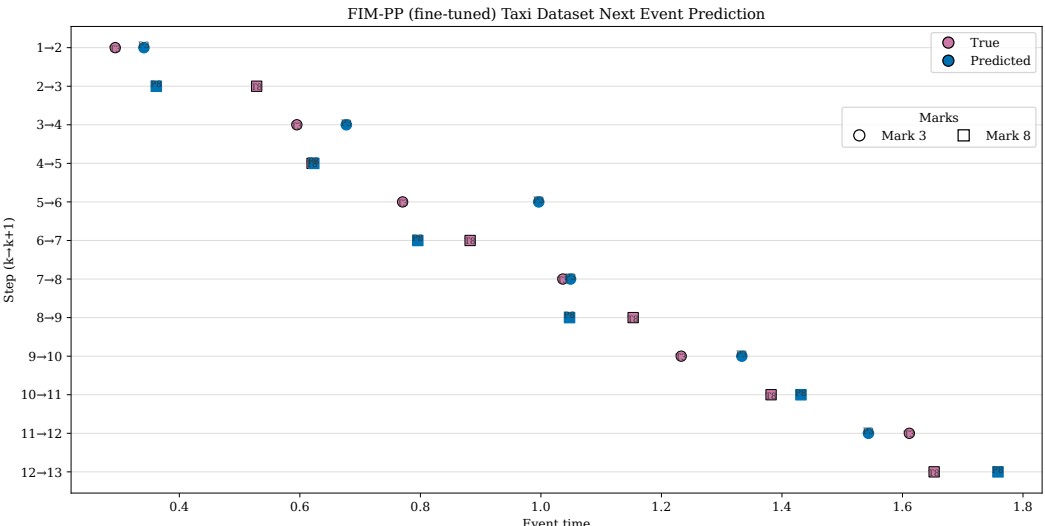

Figure 14: After fine-tuning, `FIM-PP` is able to spot the alternating pattern between mark 3 and mark 8 in the Taxi dataset.

Table 3: Prediction of $N = 20$ events in test sequences of five real-world datasets. Error-bars indicate the standard deviation over 10 trials. Results for the baseline methods were extracted from Zeng et al. (2024). Best results are bold.

| Dataset | Method | OTD | $RMSE_e$ | $RMSE_{\Delta t}$ | $sMAPE_{\Delta t}$ |
|---|---|---|---|---|---|
| **TAXI** | HYPRO | $21.653_{\pm0.163}$ | $1.231_{\pm0.015}$ | $0.372_{\pm0.004}$ | $93.803_{\pm0.454}$ |
| | Dual-TPP | $24.483_{\pm0.383}$ | $1.353_{\pm0.037}$ | $0.402_{\pm0.006}$ | $95.211_{\pm0.187}$ |
| | A-NHP | $24.762_{\pm0.217}$ | $1.276_{\pm0.015}$ | $0.430_{\pm0.003}$ | $97.388_{\pm0.381}$ |
| | NHP | $25.114_{\pm0.268}$ | $1.297_{\pm0.019}$ | $0.399_{\pm0.040}$ | $96.459_{\pm0.521}$ |
| | IFTPP | $24.053_{\pm0.609}$ | $1.364_{\pm0.032}$ | $0.384_{\pm0.005}$ | $95.719_{\pm0.779}$ |
| | TCDDM | $22.148_{\pm0.529}$ | $1.309_{\pm0.030}$ | $0.382_{\pm0.019}$ | $90.596_{\pm0.574}$ |
| | CDiff | $21.013_{\pm0.158}$ | $1.131_{\pm0.017}$ | $0.351_{\pm0.004}$ | $87.993_{\pm0.178}$ |
| | **FIM-PP (zs) (NLL)** | $23.145_{\pm0.073}$ | $1.421_{\pm0.014}$ | $\mathbf{0.277}_{\pm0.000}$ | $\mathbf{76.765}_{\pm0.386}$ |
| | **FIM-PP (zs) (S-MAPE)** | $41.543_{\pm0.026}$ | $3.986_{\pm0.009}$ | $0.297_{\pm0.000}$ | $90.149_{\pm0.366}$ |
| | **FIM-PP (f) (NLL)** | $17.914_{\pm0.117}$ | $\mathbf{0.705}_{\pm0.006}$ | $0.314_{\pm0.004}$ | $\mathbf{76.828}_{\pm0.549}$ |
| | **FIM-PP (f) (S-MAPE)** | $\mathbf{16.493}_{\pm0.034}$ | $0.707_{\pm0.004}$ | $\mathbf{0.278}_{\pm0.001}$ | $76.538_{\pm0.214}$ |
| **TAOBAO** | HYPRO | $\mathbf{44.336}_{\pm0.127}$ | $2.710_{\pm0.021}$ | $0.594_{\pm0.030}$ | $134.922_{\pm0.473}$ |
| | Dual-TPP | $47.324_{\pm0.541}$ | $3.237_{\pm0.049}$ | $0.871_{\pm0.005}$ | $141.687_{\pm0.431}$ |
| | A-NHP | $45.555_{\pm0.345}$ | $2.737_{\pm0.021}$ | $0.708_{\pm0.010}$ | $134.582_{\pm0.920}$ |
| | NHP | $48.131_{\pm0.297}$ | $3.355_{\pm0.030}$ | $0.837_{\pm0.009}$ | $137.644_{\pm0.764}$ |
| | IFTPP | $45.757_{\pm0.287}$ | $3.193_{\pm0.043}$ | $0.575_{\pm0.012}$ | $127.436_{\pm0.606}$ |
| | TCDDM | $45.563_{\pm0.889}$ | $2.850_{\pm0.058}$ | $0.569_{\pm0.015}$ | $126.512_{\pm0.491}$ |
| | CDiff | $44.621_{\pm0.139}$ | $2.653_{\pm0.022}$ | $\mathbf{0.551}_{\pm0.002}$ | $\mathbf{125.685}_{\pm0.151}$ |
| | **FIM-PP (zs) (NLL)** | $64.281_{\pm0.077}$ | $3.949_{\pm0.010}$ | $1.988_{\pm0.006}$ | $169.687_{\pm0.089}$ |
| | **FIM-PP (zs) (S-MAPE)** | $65.088_{\pm0.037}$ | $3.251_{\pm0.007}$ | $4.042_{\pm0.025}$ | $174.814_{\pm0.102}$ |
| | **FIM-PP (f) (NLL)** | $60.106_{\pm0.464}$ | $2.428_{\pm0.005}$ | $16.068_{\pm0.109}$ | $152.528_{\pm0.377}$ |
| | **FIM-PP (f) (S-MAPE)** | $58.900_{\pm0.436}$ | $\mathbf{2.347}_{\pm0.014}$ | $14.734_{\pm0.120}$ | $152.547_{\pm0.406}$ |
| **STACKOVERFLOW** | HYPRO | $42.359_{\pm0.170}$ | $1.140_{\pm0.014}$ | $1.554_{\pm0.010}$ | $110.988_{\pm0.559}$ |
| | Dual-TPP | $41.752_{\pm0.200}$ | $\mathbf{1.134}_{\pm0.019}$ | $1.514_{\pm0.017}$ | $117.582_{\pm0.420}$ |
| | A-NHP | $42.591_{\pm0.408}$ | $1.142_{\pm0.011}$ | $1.340_{\pm0.006}$ | $108.542_{\pm0.531}$ |
| | NHP | $43.791_{\pm0.147}$ | $1.244_{\pm0.030}$ | $1.487_{\pm0.004}$ | $116.952_{\pm0.404}$ |
| | IFTPP | $46.280_{\pm0.892}$ | $1.447_{\pm0.057}$ | $1.669_{\pm0.005}$ | $115.122_{\pm0.627}$ |
| | TCDDM | $42.128_{\pm0.591}$ | $1.467_{\pm0.014}$ | $1.315_{\pm0.004}$ | $107.659_{\pm0.934}$ |
| | CDiff | $41.245_{\pm1.400}$ | $1.141_{\pm0.007}$ | $1.199_{\pm0.006}$ | $106.175_{\pm0.340}$ |
| | **FIM-PP (zs) (NLL)** | $49.259_{\pm0.056}$ | $2.393_{\pm0.015}$ | $1.068_{\pm0.002}$ | $96.364_{\pm0.048}$ |
| | **FIM-PP (zs) (S-MAPE)** | $43.630_{\pm0.242}$ | $1.488_{\pm0.010}$ | $1.050_{\pm0.002}$ | $93.544_{\pm0.485}$ |
| | **FIM-PP (f) (NLL)** | $39.792_{\pm0.042}$ | $1.336_{\pm0.030}$ | $1.018_{\pm0.003}$ | $\mathbf{88.248}_{\pm0.189}$ |
| | **FIM-PP (f) (S-MAPE)** | $\mathbf{39.245}_{\pm0.056}$ | $1.407_{\pm0.022}$ | $\mathbf{1.008}_{\pm0.003}$ | $\mathbf{88.277}_{\pm0.231}$ |
| **AMAZON** | HYPRO | $38.613_{\pm0.536}$ | $\mathbf{2.007}_{\pm0.054}$ | $0.477_{\pm0.010}$ | $82.506_{\pm0.840}$ |
| | Dual-TPP | $42.646_{\pm0.752}$ | $2.562_{\pm0.202}$ | $0.482_{\pm0.012}$ | $86.453_{\pm2.044}$ |
| | A-NHP | $39.480_{\pm0.326}$ | $2.166_{\pm0.026}$ | $0.476_{\pm0.033}$ | $84.323_{\pm1.815}$ |
| | NHP | $42.571_{\pm0.293}$ | $2.561_{\pm0.060}$ | $0.519_{\pm0.023}$ | $92.053_{\pm1.553}$ |
| | IFTPP | $43.820_{\pm0.232}$ | $3.050_{\pm0.286}$ | $0.481_{\pm0.145}$ | $90.910_{\pm1.611}$ |
| | TCDDM | $42.245_{\pm0.174}$ | $2.998_{\pm0.115}$ | $0.476_{\pm0.111}$ | $83.826_{\pm1.508}$ |
| | CDiff | $37.728_{\pm0.199}$ | $2.091_{\pm0.163}$ | $0.464_{\pm0.086}$ | $81.987_{\pm1.905}$ |
| | **FIM-PP (zs) (NLL)** | $46.219_{\pm0.108}$ | $2.073_{\pm0.012}$ | $0.464_{\pm0.000}$ | $128.635_{\pm0.398}$ |
| | **FIM-PP (zs) (S-MAPE)** | $45.153_{\pm0.092}$ | $2.106_{\pm0.009}$ | $0.453_{\pm0.002}$ | $124.476_{\pm0.494}$ |
| | **FIM-PP (f) (NLL)** | $\mathbf{37.208}_{\pm0.098}$ | $2.030_{\pm0.019}$ | $0.366_{\pm0.001}$ | $\mathbf{81.188}_{\pm0.142}$ |
| | **FIM-PP (f) (S-MAPE)** | $37.454_{\pm0.060}$ | $2.116_{\pm0.002}$ | $\mathbf{0.361}_{\pm0.002}$ | $87.257_{\pm0.198}$ |
| **RETWEET** | HYPRO | $61.031_{\pm0.092}$ | $2.623_{\pm0.036}$ | $30.100_{\pm0.413}$ | $106.110_{\pm1.505}$ |
| | Dual-TPP | $61.095_{\pm0.101}$ | $2.679_{\pm0.026}$ | $28.914_{\pm0.300}$ | $106.900_{\pm1.293}$ |
| | A-NHP | $60.634_{\pm0.097}$ | $2.561_{\pm0.054}$ | $28.812_{\pm0.272}$ | $107.234_{\pm1.293}$ |
| | NHP | $60.953_{\pm0.079}$ | $2.651_{\pm0.045}$ | $27.130_{\pm0.224}$ | $107.075_{\pm1.398}$ |
| | IFTPP | $61.715_{\pm0.152}$ | $2.776_{\pm0.043}$ | $27.582_{\pm0.191}$ | $106.711_{\pm1.615}$ |
| | TCDDM | $60.501_{\pm0.087}$ | $2.387_{\pm0.020}$ | $27.303_{\pm0.152}$ | $106.048_{\pm0.610}$ |
| | CDiff | $60.661_{\pm0.101}$ | $\mathbf{2.293}_{\pm0.034}$ | $27.101_{\pm0.113}$ | $106.184_{\pm1.121}$ |
| | **FIM-PP (zs) (NLL)** | $60.238_{\pm0.161}$ | $4.172_{\pm0.064}$ | $24.057_{\pm0.050}$ | $99.069_{\pm0.390}$ |
| | **FIM-PP (zs) (S-MAPE)** | $59.392_{\pm0.149}$ | $4.323_{\pm0.042}$ | $24.804_{\pm0.014}$ | $106.317_{\pm0.187}$ |
| | **FIM-PP (f) (NLL)** | $59.437_{\pm0.082}$ | $2.703_{\pm0.012}$ | $21.985_{\pm0.014}$ | $\mathbf{87.585}_{\pm0.171}$ |
| | **FIM-PP (f) (S-MAPE)** | $\mathbf{59.150}_{\pm0.061}$ | $3.081_{\pm0.020}$ | $\mathbf{21.800}_{\pm0.025}$ | $87.754_{\pm0.109}$ |

Table 4: Prediction of 10 events in test sequences of five real-world datasets. Error-bars indicate the standard deviation over 10 trials. Results for the baseline methods were extracted from Zeng et al. (2024). Best results are bold.

| Dataset | Method | OTD | $\text{RMSE}_e$ | $\text{RMSE}_{\Delta t}$ | $\text{sMAPE}_{\Delta t}$ |
|---|---|---|---|---|---|
| **TAXI** | HYPRO | $11.875_{\pm 0.172}$ | $0.764_{\pm 0.008}$ | $0.363_{\pm 0.002}$ | $89.524_{\pm 0.552}$ |
| | Dual-TPP | $13.058_{\pm 0.220}$ | $0.966_{\pm 0.011}$ | $0.395_{\pm 0.003}$ | $90.812_{\pm 0.497}$ |
| | A-NHP | $12.542_{\pm 0.336}$ | $0.823_{\pm 0.007}$ | $0.376_{\pm 0.003}$ | $92.812_{\pm 0.129}$ |
| | NHP | $13.377_{\pm 0.184}$ | $0.922_{\pm 0.009}$ | $0.397_{\pm 0.005}$ | $92.182_{\pm 0.384}$ |
| | IFTPP | $12.765_{\pm 0.106}$ | $1.004_{\pm 0.013}$ | $0.383_{\pm 0.015}$ | $93.120_{\pm 0.526}$ |
| | TCDDM | $11.885_{\pm 0.149}$ | $1.121_{\pm 0.072}$ | $0.385_{\pm 0.009}$ | $90.703_{\pm 0.356}$ |
| | CDiff | $11.004_{\pm 0.191}$ | $0.785_{\pm 0.007}$ | $0.350_{\pm 0.002}$ | $90.721_{\pm 0.291}$ |
| | **FIM-PP (zs) (NLL)** | $13.820_{\pm 0.124}$ | $1.190_{\pm 0.013}$ | $0.281_{\pm 0.001}$ | $78.141_{\pm 0.414}$ |
| | **FIM-PP (zs) (S-MAPE)** | $19.425_{\pm 0.127}$ | $1.959_{\pm 0.015}$ | $0.299_{\pm 0.001}$ | $90.510_{\pm 0.267}$ |
| | **FIM-PP (f) (NLL)** | $8.336_{\pm 0.071}$ | $\mathbf{0.451}_{\pm 0.006}$ | $0.291_{\pm 0.004}$ | $\mathbf{75.366}_{\pm 0.160}$ |
| | **FIM-PP (f) (S-MAPE)** | $\mathbf{8.175}_{\pm 0.052}$ | $0.465_{\pm 0.004}$ | $\mathbf{0.275}_{\pm 0.001}$ | $\mathbf{73.529}_{\pm 0.138}$ |
| **TAOBAO** | HYPRO | $21.547_{\pm 0.138}$ | $1.527_{\pm 0.035}$ | $0.591_{\pm 0.019}$ | $133.147_{\pm 0.341}$ |
| | Dual-TPP | $23.691_{\pm 0.203}$ | $2.674_{\pm 0.032}$ | $0.873_{\pm 0.010}$ | $139.271_{\pm 0.348}$ |
| | A-NHP | $21.683_{\pm 0.215}$ | $1.514_{\pm 0.015}$ | $0.608_{\pm 0.011}$ | $135.271_{\pm 0.395}$ |
| | NHP | $24.068_{\pm 0.331}$ | $2.769_{\pm 0.033}$ | $0.855_{\pm 0.013}$ | $137.693_{\pm 0.225}$ |
| | IFTPP | $23.195_{\pm 0.039}$ | $2.429_{\pm 0.045}$ | $0.602_{\pm 0.037}$ | $127.411_{\pm 0.573}$ |
| | TCDDM | $\mathbf{21.012}_{\pm 0.520}$ | $2.598_{\pm 0.047}$ | $0.610_{\pm 0.022}$ | $132.711_{\pm 0.774}$ |
| | CDiff | $\mathbf{21.221}_{\pm 0.176}$ | $1.416_{\pm 0.024}$ | $\mathbf{0.535}_{\pm 0.016}$ | $\mathbf{126.824}_{\pm 0.366}$ |
| | **FIM-PP (zs) (NLL)** | $31.880_{\pm 0.040}$ | $2.024_{\pm 0.004}$ | $1.955_{\pm 0.011}$ | $170.278_{\pm 0.029}$ |
| | **FIM-PP (zs) (S-MAPE)** | $32.249_{\pm 0.041}$ | $1.822_{\pm 0.004}$ | $3.538_{\pm 0.039}$ | $172.683_{\pm 0.194}$ |
| | **FIM-PP (f) (NLL)** | $27.974_{\pm 0.162}$ | $\mathbf{1.325}_{\pm 0.010}$ | $14.954_{\pm 0.253}$ | $145.821_{\pm 1.120}$ |
| | **FIM-PP (f) (S-MAPE)** | $27.940_{\pm 0.075}$ | $1.358_{\pm 0.009}$ | $14.844_{\pm 0.055}$ | $153.499_{\pm 0.553}$ |
| **STACKOVERFLOW** | HYPRO | $21.062_{\pm 0.372}$ | $0.921_{\pm 0.019}$ | $1.235_{\pm 0.006}$ | $107.566_{\pm 0.218}$ |
| | Dual-TPP | $21.229_{\pm 0.394}$ | $0.936_{\pm 0.013}$ | $1.223_{\pm 0.010}$ | $107.274_{\pm 0.200}$ |
| | A-NHP | $22.019_{\pm 0.220}$ | $0.978_{\pm 0.023}$ | $1.225_{\pm 0.007}$ | $100.137_{\pm 0.167}$ |
| | NHP | $21.655_{\pm 0.314}$ | $0.970_{\pm 0.014}$ | $1.266_{\pm 0.003}$ | $108.867_{\pm 0.361}$ |
| | IFTPP | $22.339_{\pm 0.322}$ | $0.970_{\pm 0.011}$ | $1.251_{\pm 0.005}$ | $105.674_{\pm 0.337}$ |
| | TCDDM | $22.042_{\pm 0.193}$ | $1.205_{\pm 0.014}$ | $1.228_{\pm 0.010}$ | $108.111_{\pm 0.112}$ |
| | CDiff | $20.191_{\pm 0.455}$ | $0.916_{\pm 0.010}$ | $1.180_{\pm 0.003}$ | $102.367_{\pm 0.267}$ |
| | **FIM-PP (zs) (NLL)** | $23.527_{\pm 0.033}$ | $1.188_{\pm 0.005}$ | $1.039_{\pm 0.003}$ | $92.919_{\pm 0.556}$ |
| | **FIM-PP (zs) (S-MAPE)** | $21.182_{\pm 0.146}$ | $0.857_{\pm 0.010}$ | $1.047_{\pm 0.002}$ | $93.715_{\pm 0.388}$ |
| | **FIM-PP (f) (NLL)** | $\mathbf{19.938}_{\pm 0.093}$ | $0.823_{\pm 0.010}$ | $1.012_{\pm 0.004}$ | $87.503_{\pm 0.402}$ |
| | **FIM-PP (f) (S-MAPE)** | $\mathbf{19.846}_{\pm 0.152}$ | $0.832_{\pm 0.010}$ | $\mathbf{1.004}_{\pm 0.003}$ | $\mathbf{87.110}_{\pm 0.393}$ |
| **AMAZON** | HYPRO | $24.956_{\pm 0.663}$ | $1.765_{\pm 0.039}$ | $0.442_{\pm 0.015}$ | $83.401_{\pm 1.033}$ |
| | Dual-TPP | $25.929_{\pm 0.280}$ | $2.098_{\pm 0.101}$ | $0.475_{\pm 0.008}$ | $82.352_{\pm 1.285}$ |
| | A-NHP | $24.116_{\pm 0.807}$ | $1.741_{\pm 0.039}$ | $0.454_{\pm 0.014}$ | $84.323_{\pm 1.815}$ |
| | NHP | $25.730_{\pm 0.497}$ | $1.843_{\pm 0.053}$ | $0.491_{\pm 0.048}$ | $89.135_{\pm 1.092}$ |
| | IFTPP | $26.632_{\pm 0.519}$ | $1.955_{\pm 0.112}$ | $0.464_{\pm 0.066}$ | $89.305_{\pm 1.288}$ |
| | TCDDM | $25.091_{\pm 0.227}$ | $1.778_{\pm 0.090}$ | $0.448_{\pm 0.082}$ | $\mathbf{82.105}_{\pm 1.564}$ |
| | CDiff | $24.230_{\pm 0.287}$ | $1.766_{\pm 0.079}$ | $0.450_{\pm 0.049}$ | $\mathbf{82.124}_{\pm 2.094}$ |
| | **FIM-PP (zs) (NLL)** | $21.736_{\pm 0.115}$ | $1.141_{\pm 0.010}$ | $0.449_{\pm 0.002}$ | $120.894_{\pm 0.393}$ |
| | **FIM-PP (zs) (S-MAPE)** | $21.418_{\pm 0.142}$ | $1.198_{\pm 0.008}$ | $0.445_{\pm 0.002}$ | $118.944_{\pm 0.487}$ |
| | **FIM-PP (f) (NLL)** | $\mathbf{18.428}_{\pm 0.124}$ | $1.091_{\pm 0.016}$ | $0.361_{\pm 0.001}$ | $87.264_{\pm 0.323}$ |
| | **FIM-PP (f) (S-MAPE)** | $\mathbf{18.566}_{\pm 0.072}$ | $1.153_{\pm 0.007}$ | $\mathbf{0.352}_{\pm 0.002}$ | $82.555_{\pm 0.278}$ |
| **RETWEET** | HYPRO | $31.743_{\pm 0.068}$ | $1.927_{\pm 0.027}$ | $33.683_{\pm 0.245}$ | $105.073_{\pm 0.958}$ |
| | Dual-TPP | $31.652_{\pm 0.075}$ | $1.963_{\pm 0.038}$ | $28.104_{\pm 0.486}$ | $106.721_{\pm 0.774}$ |
| | A-NHP | $\mathbf{30.337}_{\pm 0.065}$ | $1.823_{\pm 0.031}$ | $26.310_{\pm 0.333}$ | $106.021_{\pm 1.011}$ |
| | NHP | $30.817_{\pm 0.090}$ | $1.713_{\pm 0.024}$ | $27.010_{\pm 0.429}$ | $107.053_{\pm 1.390}$ |
| | IFTPP | $31.974_{\pm 0.032}$ | $1.942_{\pm 0.062}$ | $28.825_{\pm 0.221}$ | $106.014_{\pm 0.633}$ |
| | TCDDM | $32.006_{\pm 0.074}$ | $1.789_{\pm 0.094}$ | $29.124_{\pm 0.405}$ | $106.738_{\pm 0.791}$ |
| | CDiff | $31.237_{\pm 0.078}$ | $1.745_{\pm 0.036}$ | $26.429_{\pm 0.201}$ | $105.767_{\pm 0.771}$ |
| | **FIM-PP (zs) (NLL)** | $31.027_{\pm 0.031}$ | $2.355_{\pm 0.032}$ | $27.085_{\pm 0.002}$ | $97.590_{\pm 0.152}$ |
| | **FIM-PP (zs) (S-MAPE)** | $30.986_{\pm 0.104}$ | $2.396_{\pm 0.012}$ | $28.305_{\pm 0.056}$ | $105.397_{\pm 0.496}$ |
| | **FIM-PP (f) (NLL)** | $30.592_{\pm 0.037}$ | $\mathbf{1.611}_{\pm 0.031}$ | $25.021_{\pm 0.034}$ | $86.875_{\pm 0.108}$ |
| | **FIM-PP (f) (S-MAPE)** | $30.788_{\pm 0.051}$ | $1.652_{\pm 0.018}$ | $\mathbf{24.836}_{\pm 0.042}$ | $\mathbf{85.222}_{\pm 0.086}$ |

Table 5: Prediction of 5 events in test sequences of five real-world datasets. Error-bars indicate the standard deviation over 10 trials. Results for the baseline methods were extracted from Zeng et al. (2024). Best results are bold.

| Dataset | Method | OTD | $RMSE_e$ | $RMSE_{\Delta t}$ | $sMAPE_{\Delta t}$ |
|---|---|---|---|---|---|
| **TAXI** | HYPRO | $5.952_{\pm 0.126}$ | $0.500_{\pm 0.011}$ | $0.322_{\pm 0.004}$ | $85.994_{\pm 0.227}$ |
| | Dual-TPP | $7.534_{\pm 0.111}$ | $0.636_{\pm 0.009}$ | $0.340_{\pm 0.003}$ | $89.727_{\pm 0.320}$ |
| | A-NHP | $6.441_{\pm 0.090}$ | $0.682_{\pm 0.010}$ | $0.347_{\pm 0.002}$ | $89.070_{\pm 0.152}$ |
| | NHP | $7.405_{\pm 0.122}$ | $0.641_{\pm 0.013}$ | $0.351_{\pm 0.008}$ | $91.625_{\pm 0.177}$ |
| | IFTPP | $7.209_{\pm 0.184}$ | $0.608_{\pm 0.008}$ | $0.335_{\pm 0.003}$ | $90.512_{\pm 0.169}$ |
| | TCDDM | $5.877_{\pm 0.095}$ | $0.648_{\pm 0.015}$ | $0.327_{\pm 0.005}$ | $88.051_{\pm 0.240}$ |
| | CDiff | $5.966_{\pm 0.083}$ | $0.547_{\pm 0.007}$ | $0.318_{\pm 0.003}$ | $89.535_{\pm 0.294}$ |
| | **FIM-PP (zs) (NLL)** | $6.773_{\pm 0.064}$ | $0.655_{\pm 0.013}$ | $\mathbf{0.246}_{\pm 0.001}$ | $74.912_{\pm 0.793}$ |
| | **FIM-PP (zs) (SMAPE)** | $6.763_{\pm 0.018}$ | $0.654_{\pm 0.005}$ | $0.248_{\pm 0.001}$ | $76.402_{\pm 0.802}$ |
| | **FIM-PP (f) (NLL)** | $4.083_{\pm 0.032}$ | $\mathbf{0.311}_{\pm 0.007}$ | $0.250_{\pm 0.002}$ | $\mathbf{71.108}_{\pm 0.902}$ |
| | **FIM-PP (f) (SMAPE)** | $4.103_{\pm 0.029}$ | $\mathbf{0.315}_{\pm 0.005}$ | $0.248_{\pm 0.001}$ | $73.181_{\pm 0.409}$ |
| **TAOBAO** | HYPRO | $11.317_{\pm 0.111}$ | $0.817_{\pm 0.037}$ | $0.573_{\pm 0.011}$ | $133.837_{\pm 0.524}$ |
| | Dual-TPP | $13.280_{\pm 0.092}$ | $1.877_{\pm 0.014}$ | $0.691_{\pm 0.007}$ | $134.437_{\pm 0.458}$ |
| | A-NHP | $11.223_{\pm 0.145}$ | $0.873_{\pm 0.023}$ | $0.550_{\pm 0.014}$ | $132.266_{\pm 0.532}$ |
| | NHP | $11.973_{\pm 0.176}$ | $1.910_{\pm 0.031}$ | $0.712_{\pm 0.017}$ | $134.693_{\pm 0.369}$ |
| | IFTPP | $11.052_{\pm 0.108}$ | $1.941_{\pm 0.049}$ | $0.601_{\pm 0.017}$ | $126.320_{\pm 0.591}$ |
| | TCDDM | $11.609_{\pm 0.184}$ | $1.690_{\pm 0.023}$ | $0.675_{\pm 0.009}$ | $129.009_{\pm 0.923}$ |
| | CDiff | $\mathbf{10.147}_{\pm 0.140}$ | $\mathbf{0.730}_{\pm 0.019}$ | $\mathbf{0.519}_{\pm 0.008}$ | $\mathbf{124.339}_{\pm 0.322}$ |
| | **FIM-PP (zs) (NLL)** | $15.951_{\pm 0.042}$ | $1.129_{\pm 0.007}$ | $1.761_{\pm 0.013}$ | $168.299_{\pm 0.249}$ |
| | **FIM-PP (zs) (SMAPE)** | $15.955_{\pm 0.034}$ | $1.106_{\pm 0.007}$ | $1.918_{\pm 0.026}$ | $167.486_{\pm 0.199}$ |
| | **FIM-PP (f) (NLL)** | $13.173_{\pm 0.261}$ | $0.745_{\pm 0.010}$ | $14.892_{\pm 0.370}$ | $146.921_{\pm 0.858}$ |
| | **FIM-PP (f) (SMAPE)** | $13.572_{\pm 0.166}$ | $0.759_{\pm 0.005}$ | $17.384_{\pm 0.416}$ | $150.562_{\pm 0.830}$ |
| **STACKOVERFLOW** | HYPRO | $11.590_{\pm 0.186}$ | $0.586_{\pm 0.019}$ | $1.227_{\pm 0.018}$ | $109.014_{\pm 0.422}$ |
| | Dual-TPP | $11.719_{\pm 0.109}$ | $0.591_{\pm 0.026}$ | $1.296_{\pm 0.010}$ | $106.697_{\pm 0.381}$ |
| | A-NHP | $11.595_{\pm 0.197}$ | $0.575_{\pm 0.009}$ | $1.188_{\pm 0.014}$ | $105.799_{\pm 0.516}$ |
| | NHP | $11.807_{\pm 0.155}$ | $0.596_{\pm 0.015}$ | $1.261_{\pm 0.013}$ | $108.074_{\pm 0.661}$ |
| | IFTPP | $13.124_{\pm 0.174}$ | $0.702_{\pm 0.008}$ | $1.182_{\pm 0.039}$ | $108.409_{\pm 0.692}$ |
| | TCDDM | $11.410_{\pm 0.129}$ | $0.630_{\pm 0.015}$ | $1.201_{\pm 0.028}$ | $107.893_{\pm 0.942}$ |
| | CDiff | $10.735_{\pm 0.183}$ | $0.571_{\pm 0.012}$ | $1.153_{\pm 0.011}$ | $100.586_{\pm 0.299}$ |
| | **FIM-PP (zs) (NLL)** | $11.520_{\pm 0.057}$ | $0.657_{\pm 0.003}$ | $1.030_{\pm 0.001}$ | $93.296_{\pm 0.506}$ |
| | **FIM-PP (zs) (SMAPE)** | $11.334_{\pm 0.055}$ | $0.631_{\pm 0.004}$ | $1.020_{\pm 0.000}$ | $91.864_{\pm 0.443}$ |
| | **FIM-PP (f) (NLL)** | $\mathbf{10.353}_{\pm 0.051}$ | $\mathbf{0.527}_{\pm 0.004}$ | $0.990_{\pm 0.003}$ | $86.443_{\pm 0.128}$ |
| | **FIM-PP (f) (SMAPE)** | $\mathbf{10.341}_{\pm 0.013}$ | $\mathbf{0.525}_{\pm 0.007}$ | $\mathbf{0.984}_{\pm 0.003}$ | $\mathbf{86.133}_{\pm 0.212}$ |
| **AMAZON** | HYPRO | $9.552_{\pm 0.172}$ | $1.397_{\pm 0.033}$ | $0.433_{\pm 0.008}$ | $82.847_{\pm 0.748}$ |
| | Dual-TPP | $11.309_{\pm 0.093}$ | $1.742_{\pm 0.302}$ | $0.476_{\pm 0.010}$ | $86.633_{\pm 0.573}$ |
| | A-NHP | $\mathbf{9.430}_{\pm 0.131}$ | $1.117_{\pm 0.049}$ | $0.427_{\pm 0.033}$ | $83.121_{\pm 0.415}$ |
| | NHP | $11.273_{\pm 0.198}$ | $1.431_{\pm 0.024}$ | $0.501_{\pm 0.009}$ | $90.591_{\pm 0.667}$ |
| | IFTPP | $10.230_{\pm 0.224}$ | $1.663_{\pm 0.168}$ | $0.447_{\pm 0.015}$ | $88.900_{\pm 0.610}$ |
| | TCDDM | $10.557_{\pm 0.331}$ | $1.409_{\pm 0.203}$ | $0.460_{\pm 0.032}$ | $82.401_{\pm 0.810}$ |
| | CDiff | $9.478_{\pm 0.081}$ | $1.326_{\pm 0.082}$ | $0.424_{\pm 0.018}$ | $81.287_{\pm 0.994}$ |
| | **FIM-PP (zs) (NLL)** | $11.124_{\pm 0.059}$ | $\mathbf{0.736}_{\pm 0.004}$ | $0.449_{\pm 0.004}$ | $119.129_{\pm 0.746}$ |
| | **FIM-PP (zs) (SMAPE)** | $11.029_{\pm 0.033}$ | $\mathbf{0.735}_{\pm 0.005}$ | $0.445_{\pm 0.003}$ | $117.793_{\pm 0.635}$ |
| | **FIM-PP (f) (NLL)** | $10.034_{\pm 0.060}$ | $\mathbf{0.737}_{\pm 0.006}$ | $\mathbf{0.341}_{\pm 0.004}$ | $\mathbf{78.738}_{\pm 0.339}$ |
| | **FIM-PP (f) (SMAPE)** | $10.004_{\pm 0.019}$ | $\mathbf{0.732}_{\pm 0.003}$ | $\mathbf{0.343}_{\pm 0.004}$ | $79.623_{\pm 0.524}$ |
| **RETWEET** | HYPRO | $16.145_{\pm 0.096}$ | $1.105_{\pm 0.026}$ | $27.236_{\pm 0.259}$ | $103.052_{\pm 1.206}$ |
| | Dual-TPP | $16.050_{\pm 0.085}$ | $1.077_{\pm 0.027}$ | $31.493_{\pm 0.162}$ | $101.322_{\pm 1.127}$ |
| | A-NHP | $16.124_{\pm 0.089}$ | $1.058_{\pm 0.029}$ | $29.247_{\pm 0.145}$ | $105.930_{\pm 1.380}$ |
| | NHP | $15.945_{\pm 0.094}$ | $1.113_{\pm 0.040}$ | $32.367_{\pm 0.104}$ | $107.022_{\pm 1.077}$ |
| | IFTPP | $16.043_{\pm 0.222}$ | $1.313_{\pm 0.011}$ | $30.853_{\pm 0.119}$ | $106.941_{\pm 2.031}$ |
| | TCDDM | $15.874_{\pm 0.053}$ | $1.194_{\pm 0.021}$ | $28.530_{\pm 0.110}$ | $105.570_{\pm 0.940}$ |
| | CDiff | $15.858_{\pm 0.080}$ | $\mathbf{1.023}_{\pm 0.036}$ | $26.078_{\pm 0.175}$ | $106.620_{\pm 1.008}$ |
| | **FIM-PP (zs) (NLL)** | $15.747_{\pm 0.032}$ | $1.342_{\pm 0.027}$ | $28.138_{\pm 0.068}$ | $98.668_{\pm 0.794}$ |
| | **FIM-PP (zs) (SMAPE)** | $15.780_{\pm 0.017}$ | $1.331_{\pm 0.030}$ | $28.472_{\pm 0.071}$ | $100.765_{\pm 0.803}$ |
| | **FIM-PP (f) (NLL)** | $15.645_{\pm 0.020}$ | $\mathbf{1.033}_{\pm 0.034}$ | $\mathbf{25.308}_{\pm 0.135}$ | $\mathbf{83.010}_{\pm 0.278}$ |
| | **FIM-PP (f) (SMAPE)** | $\mathbf{15.585}_{\pm 0.040}$ | $\mathbf{1.012}_{\pm 0.040}$ | $\mathbf{25.413}_{\pm 0.086}$ | $84.945_{\pm 0.343}$ |

Table 6: Summary of the parametrized base intensities and interaction kernels of Hawkes processes used in our synthetic data generation. The parameters of each configuration are sampled from uniform distributions, covering a wide range of processes.

| Dataset Configuration | Base Intensity $\mu(t)$ | Interaction Kernel $\gamma(t)$ | Parameter Distributions |
|---|---|---|---|
| Constant Base & Exponential Kernel (no interactions) | Constant $\mu(t) = c_0$ | Exponential Decay $\gamma(t) = \alpha e^{-\beta t}$ $(\gamma_{ij,\, i \neq j} = 0)$ | $c_0 \sim \mathcal{U}(0.01, 1.3)$ $\alpha \sim \mathcal{U}(0.005, 1.0)$ $\beta \sim \mathcal{U}(0.001, 10.0)$ |
| Constant Base & Exponential Kernel | Constant $\mu(t) = c_0$ | Exponential Decay $\gamma(t) = \alpha e^{-\beta t}$ | $c_0 \sim \mathcal{U}(0.01, 1.3)$ $\alpha \sim \mathcal{U}(0.005, 1.0)$ $\beta \sim \mathcal{U}(0.001, 10.0)$ |
| Sinusoidal Base & Exponential Kernel | Sinusoidal $\mu(t) = A \sin(\omega(t - \gamma)) + c_0$ | Exponential Decay $\gamma(t) = \alpha e^{-\beta t}$ | $c_0 \sim \mathcal{U}(0.05, 0.15)$ $A \sim \mathcal{U}(0.0, 10.0)$ $\omega \sim \mathcal{U}(0.1, 15.0)$ $\gamma \sim \mathcal{U}(0.0, 5.0)$ $\alpha \sim \mathcal{U}(0.1, 0.6)$ $\beta \sim \mathcal{U}(0.8, 2.0)$ |
| Gamma Base & Exponential Kernel | Gamma Shape + Constant $\mu(t) = A t^p e^{-\beta_0 t} + c_0$ | Exponential Decay $\gamma(t) = \alpha e^{-\beta_1 t}$ | $c_0 \sim \mathcal{U}(0.1, 1.3)$ $A \sim \mathcal{U}(10.0, 50.0)$ $p \sim \mathcal{U}(1.0, 2.0)$ $\beta_0 \sim \mathcal{U}(1.0, 10.1)$ $\alpha \sim \mathcal{U}(0.005, 1.0)$ $\beta_1 \sim \mathcal{U}(0.001, 10.0)$ |
| Poisson Process | Constant $\mu(t) = c_0$ | Zero Kernel $\gamma(t) = 0$ | $c_0 \sim \mathcal{U}(0.01, 1.3)$ |
| Constant Base & Rayleigh Kernel | Constant $\mu(t) = c_0$ | Rayleigh $\gamma(t) = a_0 \frac{(t - t_{\text{shift}})}{a_1^2} \exp\left(-\frac{(t - t_{\text{shift}})^2}{2a_1^2}\right)$ | $c_0 \sim \mathcal{U}(0.01, 1.3)$ $a_0 \sim \mathcal{U}(0.001, 1.0)$ $a_1 \sim \mathcal{U}(0.05, 0.25)$ $t_{\text{shift}} \sim \mathcal{U}(0.0, 0.1)$ |

Table 7: For each dataset configuration, we sample Hawkes processes with varying numbers (#) of marks, sequences and events per sequence.

| # Marks | # Samples | # Sequences | # Events per Sequence |
|---|---|---|---|
| 1 | 1000 | 2000 | 100 |
| 5 | 1000 | 2000 | 100 |
| 10 | 1000 | 2000 | 100 |
| 15 | 1000 | 2000 | 100 |
| 22 | 5000 | 2000 | 100 |

