# OpenReview forum: "In-Context Learning of Temporal Point Processes with Foundation Inference Models"
_ICLR.cc/2026/Conference — ICLR 2026 Poster_

### Official Review · Reviewer_UZoj · 2025-10-27

**Soundness:** 2
**Presentation:** 3
**Contribution:** 2
**Rating:** 2
**Confidence:** 5

**Summary:**

*disclaimer* i did not use llm at all for reviewing this paper.

This paper proposes a foundational model approach for event prediction commonly modelled by marked TPPs.  The approach is modeling the conditional intensity given history context in the pretraining, specifically modeling the parameters of a modifed hawkes process intensity that inclusion excitation and inhibition effect.  Authors provides a zeroshot and fine tune version. The fine tune performs better on next event prediction on 4 real benchmarks. For multi event prediction task results are less prominent.

**Strengths:**

A few strengths include 1) a new foundational model perspective on modeling conditional intensity.  The paper can be improved by mentioning/citing papers with other angles for foundational model for marked TPPs, ( I am sure there are a few ) and highlight how theirs differ from others) 2). Presentation is very clear, easy to follow 3). I think the thoroughness of experiments is reasonable – may include a few other datasets from benchmarks such as EasyTPP.

**Weaknesses:**

A few weaknesses include 1). The expressiveness of the modeling of conditional intensity form (can you think of something more expressive? 2.) I think the fine tune version is much prominent in downstream tasks, so it kind of diminish the claim that the transfer learning approach setting one does not have to train on individual datasets. The zeroshot version is okay. For fine tuning the authors should think about computation costs as well compare to baselines. 3). The author should consider add Transformer based baseline for a fair comparision since the proposed approach is transformer based.

**Questions:**

1.	Have the authors look at other transfer learning based approach for TPP models? Whats the benefit of modeling conditional intensity  over directly predicting time and type in the transfer learning setting?

2.	For multi event prediction,  I think the current approach is still autoregressive, the FIM-PP models are less convinced to me that they will be better than CDiff / Dual -TPP models – since the latter ones are for long horinzon prediction. It will be beneficial to provide some theoretical insight?

---

> ### Author Response · Authors · 2025-11-19
>
> Thank you for your review. We appreciate the opportunity to clarify several aspects of our work.
>
> Before addressing the individual weaknesses and questions, we would like to note that, to the best of our knowledge, **no prior work has proposed a foundation-model approach for inference of marked temporal point processes** (MTPPs). If the reviewer is aware of specific prior works in this direction, we would be grateful for the references so that we can properly acknowledge them.
>
> **@W1**: Although FIM-PP uses an exponential decay parametrisation between events (Eq. 11), both the jump magnitude and the decay rate are conditioned on the context. This allows the model to approximate localised triggering functions by e.g. predicting very large decay rates. In fact, FIM-PP is trained not only on Hawkes processes with rapidly decaying kernels, but also on Hawkes processes with localised Rayleigh kernels (see Table 3). The newly added Figure 8 in Appendix H illustrates that this exponential parametrisation can accurately approximate Rayleigh kernels.
>
> Finally,  we agree that one could adopt more expressive parametrisations; however, *our choice remains interpretable and connects to representations used in prior work* (e.g., Zhang et al., 2020).
>
> **@W2**: We thank the reviewer for highlighting the importance of reporting fine-tuning efficiency. Appendix H now includes additional information on convergence behaviour and computational cost. *In summary*, fine-tuning typically completes within a few minutes, requires fewer than 11GB of GPU memory, and inference is feasible on CPU. Moreover, the new Figure 7 shows that the pretrained model converges significantly faster and achieves higher peak performance than training from scratch.
>
> **@W3**: We are unsure what the reviewer means by the absence of Transformer-based baselines. Most baselines in our experiments (A-NHP, HYPRO, CDiff, and TCDDM) are in fact Transformer-based models. We would appreciate clarification if the reviewer had a different class of Transformer baselines in mind.
>
> **@Q1**: To the best of our knowledge, there are no existing transfer-learning–based approaches specifically designed for TPP inference. If the reviewer is aware of such work, we would be glad to include it in our related-work section.
>
> Also, we chose to model the process through the conditional intensity function because of its interpretability and its established role in TPP modelling.
>
> **@Q2**: As stated in line 366, FIM-PP is indeed an autoregressive model. Nevertheless, our empirical results in Table 1 show that FIM-PP outperforms CDiff (Zeng et al. 2024) on long-horizon prediction tasks, suggesting that the autoregressive structure does not hinder performance in this setting.
>
> *References*:
>
> - Interacting Diffusion Processes for Event Sequences Forecasting, Zeng et al. ICML, 2024.
> - Self-attentive hawkes processes, Zhang et al. ICML, 2020.

---

### Official Review · Reviewer_mxko · 2025-10-31

**Soundness:** 2
**Presentation:** 3
**Contribution:** 2
**Rating:** 2
**Confidence:** 3

**Summary:**

Until now, there have been no general-purpose foundation models for temporal point processes (TPPs). This paper introduces FIM-PP, the first foundation model for TPPs. FIM-PP can perform on real-world data as zero-shot model and can also be fine-tuned quickly. The paper's main contribution is this first attempt at a foundation model for TPPs. While many foundation models exist for vision and text, none existed for this task. Quantitatively, the model shows competitive performance in a zero-shot setting and outperforms the baselines when fine-tuned (Fig 3 and Table 1).

**Strengths:**

- As the authors note, this work is the first to propose a foundation model for temporal point processes (TPPs), thereby introducing a new and challenging task that sets an important direction for future research.
- The paper is well-organized and clearly written, making it easy for readers to follow the proposed methodology and understand the results.
- The foundation model demonstrates strong performance in estimating intensity functions - event time predictions (Table 1).

**Weaknesses:**

1. The claim of establishing a "foundation model" for Temporal Point Processes (TPPs) is not fully substantiated by the experimental evaluation. A core capability of any foundation model is its versatility across diverse data types and domains. The evaluation is limited to five datasets (Taxi, Taobao, StackOverflow, Amazon, Retweet), which primarily cover only two broad domains: Social/E-commerce and Transportation. To truly validate the model's generalized knowledge, the evaluation should be expanded to include critical TPP application domains such as Healthcare (MIMIC, EHR) and Finance (Financial Transactions), as noted in the TPP literature (e.g., [1], [2], [3]). The current limited domain coverage makes it difficult to definitively assess the model's claimed generalizability.

2. The model exhibits a noticeable weakness in event type prediction, which is a critical measure for TPP models. Specifically, the results on the Taxi dataset show a significant performance gap in event type prediction. The fine-tuned model's performance (e.g., 0.69) is notably lower than that of established baselines (e.g., 0.91). While strong time prediction performance is valuable, the poor type prediction suggests the model struggles to accurately capture the inter-dependencies between different event marks, which is essential for complex marked TPPs.

3. A key argument for using a foundation model is its rapid adaptability to new, specific tasks, as highlighted in the abstract ("rapidly fine-tuned to target systems"). The paper misses a quantitative analysis of this fine-tuning efficiency. Metrics such as the time or computational resources required for fine-tuning (e.g., convergence speed, epochs to reach peak performance) should be presented and compared against baselines to validate the model's advantage in rapid task adaptation.

[1] Shchur, Oleksandr, Marin Biloš, and Stephan Günnemann. "Intensity-free learning of temporal point processes." ICLR (2020).

[2] Mei, Hongyuan, and Jason M. Eisner. "The neural hawkes process: A neurally self-modulating multivariate point process." NIPS (2017).

[3] Liu, Zefang, and Yinzhu Quan. "Tpp-llm: Modeling temporal point processes by efficiently fine-tuning large language models." ICLR 2025 Workshop on Foundation Models in the Wild.

**Questions:**

1. Regarding the first weakness, the use of only five datasets may limit the ability to fully evaluate the foundation model’s generalization capability. Does the model perform well across diverse domains and dataset characteristics? If possible, it would be helpful to see the model’s performance evaluated on datasets with varying domains and dynamics.
2. Concerning the statement, “In contrast to some other neural methods, … we design our model to parametrize TPPs per mark conditioned on the joined history of all marks,” is this modeling approach original to your work?
3. With respect to fine-tuning the foundation model, how quickly can the reported performance be achieved?

---

> ### Author Response · Authors · 2025-11-19
>
> Thank you for the detailed and constructive review. We appreciate the reviewer’s engagement with the work and the opportunity to clarify several points.
>
> **@W1 / Q1**: We would like to clarify that our choice of evaluation datasets follows common practice in recent TPP literature, including Xue et al. (2024), Zeng et al. (2024), and Panos (2024), all of which rely on the same benchmark suite. As discussed in Appendix A, our synthetic pretraining distribution was not tailored to these datasets. Therefore, the empirical results already demonstrate meaningful generalisation to target domains that differ substantially from the pretraining prior.
>
> Expanding evaluation to additional domains such as healthcare and finance is an important direction, and we plan to explore this in future work.
>
> **@W2**: Tables 1 and 5–7 show that FIM-PP performs competitively on event-type prediction for the $OTD$ and $RMSE_e$​ metrics at horizons of 5, 10 ad 20 events, with the fine-tuned model often outperforming baseline methods. The reviewer is correct that next-event type prediction is weaker on certain datasets. We explicitly acknowledge this in the paper (lines 449–464) and explain that these particular real-world datasets display dynamics that are significantly out of distribution with respect to our current synthetic prior, which primarily affects event-type estimation.
>
> **@W3**: We thank the reviewer for this valuable suggestion. Appendix H now includes additional details on convergence speed and computational requirements. The newly added Figure 7 further shows that the pretrained FIM-PP converges substantially faster and reaches a higher peak performance when fine-tuned.
>
> In summary, fine-tuning is typically completed within a few minutes, requires fewer than 11GB of GPU memory, and inference is feasible even on CPU. The model therefore adapts quickly in practice, consistent with our claim in the abstract.
>
> **@Q1**: Please see our response to Weakness 1, which addresses this point directly.
>
> **@Q2**: No, the idea of parametrising the per-mark intensity conditioned on the joint history is not unique to our work, and we thank the reviewer for noticing the ambiguous phrasing. We will revise the text for clarity in the final version.
>
> **@Q3**: Please refer to our response to Weakness 3, where we provide fine-tuning convergence results and an updated analysis of training efficiency.
>
> *References*:
>
> - EasyTpp: Towards Open Benchmarking Temporal Point Processes, Xue et al. (ICLR 2024)
> - Interacting Diffusion Processes for Event Sequences Forecasting, Zeng et al. (ICML 2024)
> - Decomposable Transformer Point Processes, Panos (NeurIPS 2024)

---

> > ### Comment · Reviewer_mxko · 2025-11-26
> >
> > Thank you for the clear clarifications. However, I still have some reservations regarding the strength of the “foundation model” claim. I appreciate the authors’ efforts to design a foundation-style model for TPPs—this is valuable work—but I remain unsure how fundamentally new the contribution is within the broader landscape.
> >
> > That said, I acknowledge the practical value of training and having such a model, and I appreciate the authors’ efforts in this direction. From this perspective, it is somewhat disappointing that the pretraining prior does not appear sufficiently broad or general, as reflected in the weaker event-type prediction performance.
> >
> > Overall, I appreciate the authors’ responses and will raise my score.

---

### Official Review · Reviewer_VHQu · 2025-11-01

**Soundness:** 2
**Presentation:** 3
**Contribution:** 2
**Rating:** 6
**Confidence:** 4

**Summary:**

The paper proposes FIM-PP, a marked temporal point process foundation model that extends the foundational inference model (FIM) pretraining paradigm. Specifically, FIM-PP simulates synthetic data from several predefined parametric intensity functions and leverages an autoregressive attention-based encoder-decoder framework, enabling in-context learning on novel datasets to estimate the parameters of a Hawkes process with exponential decay. Experimental results over long time horizons demonstrate that the proposed approach achieves competitive zero-shot and fine-tuned event prediction performance on four datasets, with mixed results for next-event prediction compared to dataset-specific baselines on two datasets.

**Strengths:**

- The paper is well written and easy to follow.
- The adaptation of FIM to marked temporal point processes using an autoregressive attention-based encoder-decoder framework that enables in-context learning is both interesting and novel.
- Experimental results over long time horizons demonstrate that the proposed approach achieves competitive zero-shot and fine-tuned event prediction performance on four datasets.
- The paper provides both the model and data, enabling reproducibility.

**Weaknesses:**

- The performance on novel datasets appears to depend heavily on the quality and diversity of the generated synthetic data. A focused discussion on how the synthetic data is generated, namely, which parametric intensity functions are assumed would be valuable. Additionally, it would strengthen the submission to include an experimental or theoretical analysis quantifying uncertainty at inference time, particularly in relation to the similarity between inference dataset and the synthetic data used during training, similar to approaches in Gaussian processes.

- FIM-PP consistently estimates the parameters of a Hawkes process with an exponential decay kernel for any novel dataset. However, it is unclear how this approach would generalize to other triggering kernels, especially those with localized effects. The paper should also discuss and include benchmarks against methods such as [1,2], which assume more flexible triggering kernels for Hawkes processes.

- The proposed approach appears computationally intensive compared to dataset-specific methods, especially given the mixed performance gains observed in zero-shot tasks. The paper should provide a detailed discussion of the computational complexity of the proposed method, including the number of parameters, memory requirements, training and inference time, and how the model scales with the number of marks. It is also unclear why the number of marks is capped at 22, particularly since many clinical MTPP datasets, such as MIMIC, contain more than 22 marks.

- The results for next-event prediction, when compared to dataset-specific baselines on two datasets, are mixed and warrant further discussion.

- The proposed approach does not explore transferring the learned FIM-PP representations to alternative dataset-specific models for initialization, which could enable the incorporation of dataset-specific modeling assumptions. Exploring this aspect would strengthen the submission.

**References**
- [1] Isik Yamac et al., "Hawkes Process with Flexible Triggering Kernels", MLHC  2023.
- [2] Pan Zhimeng et al., "Self-adaptable point processes with nonparametric time decays", NeurIPS 2021.

**Questions:**

- Are all model parameters retrained during fine-tuning of FIM-PP?
- Could you clarify why FIM-PP achieves mixed next-event prediction results compared to dataset-specific baselines? Additionally, how does the model's performance vary as the prediction time window increases beyond N=1? What is the largest value of N for which the model can still predict with reasonable performance?
- Table 2: Could you provide complete results for all baselines across all datasets?
- Are the representations learned by FIM-PP transferable to alternative dataset-specific models for initialization, in order to support dataset-specific modeling assumptions?

---

> ### Author Response · Authors · 2025-11-19
>
> Thank you for the detailed and thoughtful review. We appreciate the reviewer’s careful reading and constructive suggestions, which helped us clarify several aspects of the paper.
>
> **@W1**: Appendix A of the original submission contains the full specification of our synthetic Hawkes prior; in particular, Table 3 details the families of base intensities and kernels used during pretraining. We agree that these choices should be emphasized more clearly in the main text, and we will highlight them more prominently in the final version.
>
>  **@Uncertainty estimation**: we fully share the reviewer’s interest; this is an important topic in Bayesian inference, especially for zero-shot models. However, we consider such an analysis beyond the scope of this first proof-of-concept work, whose primary goal is to demonstrate that a *foundation inference model can indeed be pretrained for zero-shot inference of MTPPs*. We view uncertainty quantification as a natural direction for future work.
>
> **@W2**: Although FIM-PP uses an exponential decay parametrisation between events (Eq. 11), both the jump magnitude and the decay rate are context-dependent, allowing the model to approximate localised triggering functions by e.g. predicting large decay rates. Importantly, FIM-PP is trained not only on Hawkes processes with fast exponential decay, but also on processes with localised Rayleigh kernels (see Table 3).
>
> The new Figure 8 illustrates that this parametrisation can closely approximate Rayleigh kernels and even power-law kernels, despite the latter being absent from the training distribution. Regarding the additional Benchmarks, we will include the suggested references in our related-work section.
>
> **@W3**: We thank the reviewer for raising this point. Appendix H now includes additional information on convergence speed and hardware requirements. To summarise: as stated in Appendix C.2, the model contains 16.1M parameters. Fine-tuning requires fewer than 11GB of GPU memory and typically completes in a few minutes; inference is even feasible on CPU.
>
> The complexity with respect to the number of marks is linear because the model outputs one parametrisation per mark, although in practice the cost is considerably lower due to caching of context embeddings.
>
> Finally, pretraining requires specifying the maximum number of marks; we chose 22 since this covers all datasets used in recent neural point process benchmarks. Beyond computational considerations, nothing fundamentally limits FIM-PP from supporting more marks, and we plan to explore this direction in future work.
>
> **@W4**: We analysed the mixed next-event prediction results in Section 5.4 and Appendix G. At present, we do not have a definitive explanation for the performance variability of the pretrained model across certain datasets.
>
> **@W5**: We may have misunderstood the reviewer’s intention here. If the suggestion is to transfer hidden representations from FIM-PP to initialise dataset-specific models, **we did not pursue this direction in the present paper**, though we agree it is a promising avenue for future work. We note, however, that *FIM-PP is explicitly transferring knowledge from the synthetic Hawkes prior to target datasets that are, by construction, out-of-distribution*.
>
> **@Q1**: Yes. With only 16.1M parameters  (Appendix C.2), fine-tuning all weights is entirely feasible and completes in a few minutes on a single GPU. We will clarify in the final version of the manuscript that all parameters are updated during fine-tuning. Future work may investigate lighter fine-tuning methods such as LoRA.
>
> **@Q2**: As shown in Figure 3(a) and in Tables 5–7 of the Appendix, FIM-PP performs well for horizons $N=(5, 10, 20)$. We did not evaluate horizons beyond 20 due to a lack of baselines for comparison, but there is no inherent architectural limitation preventing the model from predicting further than $N=20$.
>
> **@Q3**: The experimental setup and reported next-event baselines in Table 2 follow Zeng et al. (2024), who only evaluated these two datasets and did not report all baselines for next-event prediction.
>
> **@Q4**: We did not explore this transfer-learning direction in this initial work. Nevertheless, one could plausibly build on the history-and-context representation in Eq. 9 to initialise dataset-specific models. We thank the reviewer for the suggestion, which we will mention as a promising future extension.
>
> *References:*
> - Interacting Diffusion Processes for Event Sequences Forecasting, Zeng et al. ICML, 2024.

---

### Official Review · Reviewer_BVzB · 2025-11-02

**Soundness:** 2
**Presentation:** 3
**Contribution:** 3
**Rating:** 6
**Confidence:** 3

**Summary:**

This paper introduces FIM-PP (Foundation Inference Model for Point Processes), a Transformer-based model for inferring conditional intensity functions of Marked Temporal Point Processes (MTPPs) through in-context learning. Instead of training a dedicated model for each event dataset, the authors pretrain FIM-PP on a large synthetic corpus of Hawkes processes, learning to infer conditional intensities directly from event histories and contextual sequences. Once pretrained, the model can be applied to new datasets, zero-shot or fine-tuned with minimal data.

**Strengths:**

1. The idea is interesting.

2. The paper pioneers the application of foundation inference models and in-context learning to temporal point processes—an original and timely contribution aligning with the broader movement toward general-purpose dynamical system learners.

3. The writing is good.

**Weaknesses:**

1. The pretraining prior is entirely Hawkes-based; real-world processes often deviate from this form (e.g., non-linear, periodic, or exogenous influences). The authors do not quantify or discuss this distributional gap.

2. While the concept of amortized inference for MTPPs is appealing, the paper provides little theoretical justification for why in-context learning generalizes to unseen, potentially non-Hawkes dynamics.

3. Transformer-based FIMs may struggle with long event sequences and large contexts. The paper lacks a complexity or runtime analysis.

**Questions:**

1. How does FIM-PP perform on non-Hawkes synthetic processes (e.g., power-law kernels or renewal processes)?

2. What is the sensitivity of zero-shot performance to the number of context sequences provided?

---

> ### Author Response · Authors · 2025-11-19
>
> We thank the reviewer for the thoughtful and constructive review.
>
> **@W1**: As stated in the Appendix and Table 3, our model is indeed pretrained on synthetic data sampled from Hawkes processes **with time-dependent base intensity functions**. We fully agree that real-world processes often deviate from this assumption. However, our Hawkes prior already includes non-linear behaviour (max function) that allows for inhibitory events, as well as diverse exogenous drives implemented through time-dependent base intensities. These include periodic signals and localized transient signals (Gamma-shaped). See Table 3 for details.
>
> **@Distributional gap**: In the newly added Appendix H, we now provide detailed statistics of both the synthetic pretraining corpus and all target datasets. Figures 9–14 show that the Hawkes-based prior is very broad and partially overlaps with the empirical distributions of all target datasets. From a Bayesian perspective, this overlap helps explain why FIM-PP can infer posterior intensities that meaningfully capture the dynamics of the target data. We will expand this argument in Section 4.1 of the final version.
>
> **@W2**: This point is closely related to W1, and we hope that our analysis based on prior–target overlap addresses this concern.
>
> **@W3**: We agree, but note that this limitation is shared by all attention-based architectures rather than being specific to FIM-PP. In practice, longer contexts can be handled by partitioning the context set into chunks of size $K$ (say $K=2000$) and averaging the inferred intensities across chunks, resulting in linear asymptotic complexity.
>
> **@Q1**: The newly added Figure 8 reports zero-shot performance on (i) Hawkes processes with power-law kernels (explicitly out-of-distribution) and (ii) renewal processes with exponential inter-event gaps (in-distribution). In both settings the model recovers the target intensities well. We also emphasize that all real-world datasets used in our experiments (Tables 1 and 2) are, of course, non-Hawkes.
>
> **@Q2**: We appreciate this question. The updated Figure 6 shows that FIM-PP’s zero-shot performance is robust with respect to the number of context sequences provided.

---

### Author Response · Authors · 2025-11-19

We thank all reviewers for their thoughtful and constructive feedback.

We would like to emphasise that the primary contribution of this work is not the introduction of yet another neural point-process model, but the establishment of **the first amortised, simulation-based inference framework for temporal point processes** that operates fully zero-shot; a capability that, to our knowledge, does not exist in prior literature.

All existing “foundation-style” or transfer-learning approaches for TPPs, including the LLM-based method cited by reviewers (Tpp-LLM, ICLR 2025 Workshop), require dataset-specific training or fine-tuning. Therefore, these approaches do not amortise inference in the simulation-based sense: they do not yield a model that can be directly applied to entirely new datasets without retraining.

*In contrast*, FIM-PP is pretrained once on a broad synthetic prior and can be applied immediately to unseen MTPP datasets (including fully out-of-distribution cases such as non-Hawkes processes with power-law kernels, which we now include in the Appendix).

The fine-tuning experiments in the paper are intentionally optional, meant to show that *the pretrained model provides an excellent initialization*, while the core conceptual contribution is the amortized zero-shot capability itself.

We hope this framing clarifies that considerations such as runtime comparisons or dataset-specific tuning, while relevant, should not overshadow the main message: FIM-PP introduces a genuinely new inference paradigm for temporal point processes, analogous to the shift from per-task training to pretrained models observed in other domains.

---

### Meta-Review · Area_Chair_sCDc · 2026-01-13

**Summary:**

I’m leaning accept. The paper proposes a new amortized, simulation-based inference paradigm for marked TPPs. You pretrain once on a broad synthetic prior, then infer intensities in-context on new datasets, rather than about squeezing another point-process model into the benchmark zoo. Such (pre-fitted network) paradigm has been explored both in tabular and time series. The reviews converge on “interesting, clearly written, and direction-setting,” but they also converge on one core risk: the current evidence may not fully justify the “foundation model” label, given a Hawkes-centric pretraining prior, uneven event-type performance, and initially underdeveloped compute/scaling discussion. The rebuttal meaningfully strengthens the empirical and systems narrative, while overall I second the concern on limited radius to claim "foundation," I still think it's valuable and meaningful step forward. I’d want the final version to sharpen the claim (foundation for inference, not universal domain coverage yet), keep the new analyses in the main paper (not buried), and be explicit about failure modes (especially mark prediction).

Aside from the paper itself, I'd like to call out their repo (https://fim4science.github.io/OpenFIM/intro.html) contains author identities, violating the anonymous policy.

**Reviewer Concerns:**

The rebuttal to a large extent, addresses the biggest technical critiques. It clarifies what’s in the synthetic prior, adds prior–target statistics/overlap analysis, provides additional OOD synthetic results, and quantifies fine-tuning cost/footprint with concrete numbers. What remains is mostly “scope” and “claim calibration”: limited domain diversity in real-data evaluation (healthcare/finance), lack of uncertainty quantification for zero-shot deployment, and no clean resolution of the mixed next-event results and type-prediction gaps beyond “OOD to our current prior.”

**Reviewer Scores:**

I'd expect two positive reviews stay that way, one raising their score (as they explicitly indicates) from 2 to possible 4, and one last reviewer likely maintaining a score of 2.

---

### Decision · Program_Chairs · 2026-01-26

Accept (Poster)